



# Autonomous Airborne Mid-IR Spectrometer for High Precision Measurements of Ethane during the NASA ACT-America Studies

Petter Weibring[1]\*, Dirk Richter[1], James G. Walega[1], Alan Fried[1], Joshua DiGangi[2], Hannah Halliday[2], Yonghoon Choi[3], Bianca Baier[4,5], Colm Sweeney[4,5], Ben Miller[4,5], Kenneth J. Davis[6], Zachary Barkley[6]
and Michael D. Obland[2]

[1] Institute of Arctic and Alpine Research, University of Colorado, Boulder, CO, USA
[2] NASA Langley Research Center, Hampton, VA, USA
[3] Science Systems and Applications Inc., Hampton, VA, USA
[4] Cooperative Institute for Research in Environmental Sciences, University of Colorado, Boulder, CO, USA
[5] NOAA ESRL Global Monitoring Division, Boulder, CO, USA
[6] Department of Meteorology & Atmospheric Science, The Pennsylvania State University, University Park, PA, USA

Correspondence to: Petter Weibring (petter.weibring@colorado.edu)

**Abstract**

An airborne trace gas sensor based on mid-infrared technology is presented for fast (1-second) and high precision ethane
measurements during the Atmospheric Carbon and Transport-America (ACT-America) study. The ACT-America campaign
is a multi-year effort to better understand and quantify sources and sinks for the two major greenhouse gases carbon dioxide
and methane. Simultaneous airborne ethane and methane measurements provide one method by which sources of methane
can be identified and quantified. The instrument described herein was operated on NASA's B200 King Air airplane spanning
five separate field deployments. As this platform has limited payload capabilities, considerable effort was devoted to
minimizing instrument weight and size without sacrificing airborne ethane measurement performance. This paper describes
the numerous features designed to achieve these goals. Two of the key instrument features that were realized were
autonomous instrument control with no on-board operator and the implementation of direct absorption spectroscopy based
on fundamental first principles. We present airborne measurement performance for ethane based upon the precisions of zero
air background measurements as well as ambient precision during quiescent stable periods. The airborne performance was
improved with each successive deployment phase, and we summarize the major upgraded design features to achieve these
improvements. During the 4th deployment phase, in the spring of 2018, the instrument achieved 1-second (1σ) airborne
ethane precisions reproducibly in the 30 - 40 parts-per-trillion by volume (pptv) range in both the boundary layer and the less
turbulent free troposphere. This performance is among some of the best reported to date for fast (1 Hz) airborne ethane
measurements. In both the laboratory conditions and at times during calm and level airborne operation these precisions were
as low as 15 - 20 pptv.



## 1 Introduction

The Atmospheric Carbon and Transport-America (ACT-America) campaign was a four year study composed of five different aircraft campaigns over the continental U.S. to quantify sources, sinks and transport of carbon dioxide ($CO_2$) and methane ($CH_4$), two of the major greenhouse gases. There are a multitude of sources of methane emitting into the
atmosphere, such as: oil & natural gas exploration and production (i.e., emissions from drilling, on-site processing and storage, flaring, transmission, etc.), coal mines, wildfires, as well as from biogenic emissions from ruminants and associated manure, landfills, water treatment plants, wetlands, and stagnant water ponds, to name a few. In order to evaluate their respective contribution of total emissions, it is important to distinguish and quantify these various sources. One method that has successfully been employed is to utilize fast simultaneous measurements of $CH_4$ with ethane ($C_2H_6$). Both gases are co-
emitted from oil & natural gas production in varying amounts depending upon the particular shale formation and specific production activity. Fast measurements, precisely co-aligned in time to remove temporal instrument differences, results in highly correlated emission ratios. By contrast, biogenic methane sources reveal enhanced methane with no enhancements in ethane. In addition to its role in characterizing methane sources, ethane is the longest-lived and most abundant non-methane hydrocarbon and since its reaction rate with OH is ~ 40 times higher than methane-OH at 298-K, large enhancements in
ethane relative to methane can dramatically affect OH levels, and hence ethane acts as an indirect Greenhouse gas. This paper discusses the development and deployment of a precise, accurate, and fast instrument that can reliably measure ethane on small low-flying aircraft and provide invaluable information related to Greenhouse emissions.

Richter et al. (2015) discuss the precursor of the instrument presented here for high performance airborne measurements of
ethane coupled with simultaneous measurements of formaldehyde (CAMS-1:Compact Atmospheric Multi-Species Spectrometer). CAMS-1 employs a tunable mid-IR laser source based upon difference frequency generation (DFG) to access strong vibrational-rotational lines in the mid-IR spectral region. Richter et al. (2015) and Weibring et al. (2006, 2007, 2010) discuss the performance advantages of DFG based technology for this purpose. Measuring formaldehyde and ethane simultaneously, CAMS-1 achieved a 1-second (1σ) airborne precision of 40 - 50 parts-per-trillion by volume (pptv) and 15 -
20 pptv, respectively, for formaldehyde and ethane. All ethane precisions discussed in this paper refer to 1-second 1σ levels. However, CAMS-1 is too large and too heavy for operations on the NASA B200 King Air turboprop aircraft employed during ACT-America (requires 2 large aircraft racks and weighs between 600 and 700 pounds, depending upon the exact configuration). CAMS-1, furthermore, requires an onboard operator, which adds an additional weight of approximately 250 - 300 pounds (operator and seat). For the ACT-America study, an instrument with the performance of CAMS-1 was needed to
satisfy the limited space, power, weight capabilities, and ability to accommodate on-board operators provided by the aircraft selected to carry out the study. Aside from the larger platforms (e.g. NASA DC-8, NCAR C130), smaller airborne platforms are being increasingly utilized as more flexible and economic platforms to study atmospheric science questions. CAMS-2 was designed to be easily accommodated by these platforms.



Yacovitch et al. (2014), Smith et al. (2015), and most recently Kostinek et al. (2019) reported the use of a smaller and lighter weight high performance IR laser system from Aerodyne, Inc. and successfully recorded high quality and fast ethane measurements. The latter paper describes improvements to such systems for high performance measurements of $CH_4$, $CO_2$, CO, $N_2O$ in addition to $C_2H_6$ on the NASA C-130 aircraft during ACT-America. Both the C-130 and B200 were deployed with similar payloads and coordinated flight paths to study the transport of greenhouse gases by mid latitude weather

systems e.g. Pal et al., (2020), quantification of regional, season fluxes of $CO_2$ (Feng et al., 2019; and Zhou et al., 2020) and $CH_4$ Barkley et al, (2019a,b), and evaluation of the Orbiting Carbon Observatory-2 (OCO-2) satellite Bell et al., (in press). Typical airborne ethane measurement precisions reported by Yacovitch et al. (2014) and Smith et al. (2015) on average fell in the 80 pptv range, which is about a factor of 4 higher than when the aircraft was on the ground. Kostinek et al. (2019) further break out their airborne measurement precisions for both the free troposphere, where the effects of aircraft turbulence

and vibrations are minimal, and in the planetary boundary layer (PBL) where the opposite is the case. They report ethane precisions of 146 pptv in the free troposphere (smooth flight conditions) and 205 pptv in the PBL (frequent turbulence). Kostinek et al. (2019) and references therein, also discuss the fact that airborne measurement precisions of these spectrometers are dramatically affected by cabin pressure changes as the aircraft ascends and descends to different flight levels or altitudes. To address this, these researchers carried out frequent addition of calibration standards every 5 – 10

minutes for a total duration of 20-seconds, which includes a 10-seconds flush time. As shown by Kostinek et al. (2019) this procedure minimized in-flight discrepancies compared to measurements of methane carried out with a separate cavity ring down based spectrometer.

The effects of cabin pressure changes on retrieved mixing ratios is not unique to Aerodyne spectrometers, and have also been

observed with our wide variety of previous IR instruments in past airborne deployments. The cabin pressure effect is endemic to all such spectrometers without optical compartment pressure control. Pressure perturbations can cause multiple effects such as: movement of optical fringes in the open-air path external to the sample cell; changes in background baseline features from deflection of windows and other components as well as changes in analyte concentrations in the open-air path, and other effects specific to the optical measurement configuration. Small differences in the optical structure between

measurements and instrument background/zeroing imposes a time-dependence on the effects of such pressure changes, which may or may not be reproducible with pressure. For the detection of molecular species with smaller absorption cross sections and/or smaller atmospheric concentration at the ppbv or pptv level, such technical noise often fundamentally limits the quality of measurement and scientific value. To mitigate this effect, CAMS-2 employed a pressure-stabilized enclosure around the entire optical system.


Like its predecessor, CAMS-2 employs a mid-IR laser source based upon Difference Frequency Generation (DFG) technology. We discuss herein the numerous designs implemented to reduce weight and size and to incorporate autonomous



instrument control without the need for an on-board operator. This system reliably acquired high precision and fast ethane measurements (30-40 pptv) on the B200 aircraft over several hundred flight hours during the 1st – 4th ACT-America

deployment phases. The airborne performance was improved with each successive field deployment phase study, and we summarize the major upgrades to achieve these improvements. We also show that the retrieved ethane background values surrounding each ambient period can be used to estimate one component of the total measurement uncertainty (TMU), as will be discussed. We also present comparisons with NOAA/ESRL's Global Monitoring Division programmable flask package (PFP) ethane measurements acquired on the same aircraft and show example correlations with methane in providing

methane source characterizations.

## 2 Instrument Design and Set-up

The instrument is mounted to a Welch (Welch Mechanical Design, LLC) rack (33.6 in length x 24 in depth x 20.2 in width, 45 lbs) and consists of several sub-assemblies. The laser spectrometer and a data acquisition system are mounted inside a temperature-, pressure- and vibration-controlled vessel mounted to the top of the rack, while a gas flow control and

calibration system, including a vacuum pump, and an uninterruptible power system (UPS) are mounted to the interior of the rack. Figure 1 shows a photo of the instrument with the major system components as deployed in the cabin of the NASA King Air B200 aircraft.

### 2.1 DFG Laser Source & Detection Module

The spectrometer consists of three parts: 1) the seed lasers and fiber amplifiers; 2) the DFG mid-IR generation and detection

module; and 3) the multipass sampling cell. These are all shown in Fig. 2. The laser module is based on two fiber-coupled diode laser sources and fiber amplifiers, which are mounted on a vibration-damped base plate inside the spectrometer enclosure.

Both the signal laser (1562 nm Distributed Feedback) and pump laser (1063.5 nm Distributed Bragg Reflector), are

computer controlled for wavelength scanning. The laser outputs are amplified in custom built rare-earth-doped erbium (Er) and ytterbium (Yb) fiber amplifiers and produce up to 500 mW and 800 mW of optical output power. The fiber outputs are fusion spliced to a wavelength division multiplexer (WDM). The fiber gain sections are backward pumped by Bragg grating stabilized diode lasers (976 nm). Faraday optical isolators are used to minimize optical feedback to the seed lasers and fiber amplifier gain section. The combined fiber amplifier outputs are focused into a 1 mm thick and 50 mm long non-linear

periodically poled lithium niobate (PPLN) crystal to generate tunable mid-IR radiation. The signal (MFD=9.5 μm) and pump (MFD=6.2 μm) beams are imaged (M=18) into the PPLN crystal with a two lens system consisting of a f=2.75 mm aspheric lens (L1 in Fig.2)  followed by a plano-convex f=50 mm CaF$_2$ lens (L2 in Fig. 2). The PPLN crystal is mounted to a copper block attached to a Peltier element and is heated to a temperature of about 40 C to satisfy the phase-matching condition. To



maximize the conversion process in the PPLN crystal, the polarization of the individual signal lasers are adjusted to a linear polarization state by in-line polarization controllers (not shown). As shown in Fig. 2, the converted mid-IR idler beam at the output of the PPLN is imaged by a $CaF_2$ lens (L3, 50 mm) into the multi-pass absorption cell (MP) configured for an effective optical path length of 47.6-m. The remaining unconverted signal and pump radiation exiting the PPLN are removed by a Germanium filter (F) and reflections off this filter are directed onto a series of absorbent glass filters (not shown). The PPLN module is shielded to prevent scattered pump and signal light from reaching the detectors in the detection module. The mid-IR beam then passes through two beam splitters (S1 and S2), before being directed into the MP. The first (S1) splits off ~1% of the beam, which is directed through a cell containing pure ethane ($C_2H_6$) (0.4-torr) and onto the reference detector (RD) for computer controlled passive wavelength locking/tracking. The second beam splitter (S2), splits off 50% of the remaining beam and is then focused by a 25 mm $CaF_2$ lens onto an amplitude modulation detector (AMD). This allows close matching of the beam intensities and spectral features on the AMD and cell detector CD (L5, f=25 mm $CaF_2$ lens) to remove common-mode optical noise from the laser source assembly, including fiber optic components. Neither apertures nor special coatings were applied in the detection module housing to suppress scattered light except for a couple of beam dumps to reduce the impact of reflections originating from the immersion lens of each detector. The optical components are affixed to the baseplate by UV cured epoxy after alignment.

## 2.2 MP Cell & Opto-Mechanical Design

Similar to the patented multi-pass cell design employed in CAMS-1 (Richter et al., 2015) the present MP offers long path lengths and smaller sampling volumes (~ 1 liter) than traditional Herriott cells. This is accomplished employing a sealed hollow core tube in addition to an outer cylindrical tube that provides a vacuum-tight optical sampling cell. The inner tube is mounted centered to the cell's longitudinal optical axis, reducing the sampling volume between the two spherical mirrors of a traditional Herriott cell. Its diameter is limited to a radius that provides sufficient clearance of the recirculating beams between the two orthogonally placed spherical mirrors. In addition, this patented design (Richter et al., 2013) significantly reduces the optical scattering that is received by the detectors. A solid non-flexing opto-mechanical coupling between the DFG components and the detectors is of utmost importance, as it minimizes intensity perturbations and optical baseline shape changes. One end of the multi-pass cell is mounted solid to this base assembly, while the other end is left floating to avoid mechanical stress due to thermal expansion for when the system is not actively temperature controlled (not in use).

The core inner tube of the MP cell is made out of carbon fiber providing excellent stiffness and low thermal expansion. The MP cell spherical mirrors have an outer diameter of 63.5 mm with a centroid circular hole of 35 mm, prescribing a torus (donut) shape. The mirror is mounted to a cylindrical flange which in turn is suspended by six polished stainless steel rods connected to the end of the inner tube flange. The opto-mechanical arrangement allows the flange to slide along the rods for adjustment of the mirror separation to allow adjustment for tolerances of the MP mirror radius of curvature and obtain a


circular pattern with the desired pathlength and number of roundtrip reflections. The mirror flange also accommodates a simple tip-and-tilt design to compensate for any machining tolerances of the mirrors or angular offsets of the carbon fiber inner tube. A borosilicate glass tube is used as the outer cell body to provide visible access to trace an alignment beam. The

beam is launched from one side of the MP cell and exits the cell on the opposing side, allowing for a compact set-up with a close mounting of detectors. The entire spectrometer, including the MP cell, DFG laser source with seed lasers and fiber amplifiers, current and temperature controllers, FPGA and power supplies are arranged into a compact package that fits into a 12 inch diameter pressurized and thermally controlled enclosure. All optical fibers are embedded in memory foam to minimize the pick-up of acoustic noise and prevent the movement of the optical fibers during airborne operation.

**2.3 Electronics: Power Supplies, Detectors, Filters, Preamps, FPGA, & Communications**

For this instrument, electronics and control systems were designed to support autonomous and calibration-free operation. This included the use of low power-consumption electronic components, minimizing thermal impact, and reduced weight and size. Electronic components and circuits were designed to operate with a low electronic noise floor well beyond desired sensitivity requirements. One method to achieve significant savings in weight and size was accomplished by replacing large

and heavy linear power supplies with switching power supplies.

Desired electronic performance was achieved by employing: 1) low noise (Vpp<5 mV output) power supplies (PS) with appropriate filtering; 2) judicious design of power and grounding pathways; 3) low noise laser diode (LD) drive electronics as well as low-noise detector amplification; 4) all components controlled by a single embedded computer with synchronized

arbitrary waveform generation and data acquisition at 320 kHz; and 5) computerized signal processing, yielding an electronic noise floor corresponding to a fractional minimum absorbance of $A_{min}{\sim}1\text{-}2\text{x}10^{-6}$ for a power level of $\sim$10-20 $\mu$W. All electronic components are schematically shown in Fig. 3, and further details regarding items (4) and (5) above will be discussed in Section 2.6.

The CD, AMD and RD detectors are three-stage Peltier cooled (-60℃) Vigo HgCdTe detectors ($D^{*}{\sim}5\text{x}10^{10}$@1kHz, Rs$\sim$500k, Cs$\sim$400pF, d=0.1mm), with immersed ball lenses (d=1mm), providing almost identical response and noise characteristics. The detectors, operating in photoconductive mode, are matched to low noise trans-impedance amplifiers (TIA) directly located at the detectors, yielding a trans-impedance gain of $\sim$100x$10^3$. The TIA outputs are sent into band pass (BP) filter channels for each detector before digitization by the computer system. The CD TIA output is also sent into a low

pass (LP) filter channel to measure the transmission power of the laser through the MP cell, allowing compensation of beam path fluctuations and mirror degradations. The spectrometer computer system is based on a real-time Linux host and a Field Programmable Gate Array (FPGA) used for all IO functions. The FPGA controls the arbitrary waveform generator and the 16-bit analog to digital (AI) converter as well as custom timing and safety control of the laser drivers. The FPGA also



handles housekeeping (temperatures, pressures, flows, etc.) using a combination of AI/AO/DIO and I²C sensors.
Communication between spectrometer, flow system (see section 2.4), operator, and service technician is handled by a wi-fi router with built-in cellular modem. The operator controls the system using either an iPad or Android device. Figure 3 shows a schematic of the various electronics systems.

**2.4 Calibration, Flow & Pressure Stabilized Optical Enclosure Systems**

The gas handling system (Fig. 4) is comprised of: 1) an inlet system with a port for introduction of zero air and calibration
mixtures; 2) zero air and calibration cylinders with appropriate flow control (FC) and suck-back controllers; 3) the MP cell with inlet pressure control (PC) and outlet flow meter (FM); 4) the vacuum pump with manual shutoff and flow control values (V1 and V2); and 5) real-time Linux computer with integrated FPGA (analog-, digital- input/output).

Ambient air is sampled perpendicular to the aircraft flow through a heated stainless steel inlet (35 C 0.38-cm ID) located
outside the fuselage boundary layer, and is drawn through a heated teflon line, through a 3-micron particle filter, through a pressure controller (MKS640A) and into the MP. The total inlet length from the inlet entrance to the cell entrance is ~ 6 m. The MP cell pressure is controlled to 73 torr +/- 0.1 torr. The reduced pressure gas is fed through the pressure-stabilized vessel surrounding the entire optical system (shown in Fig. 1 as the cylinder) and into the multi-pass cell using flexible Teflon tubing to reduce vibration coupling. Similarly, all electrical connections to and from the optical system were directed
through vacuum feedthroughs. Sample flow rates of ~ 4 slm, yields cell resonance times of ~1 s (1/e) before exiting the aircraft via a flow meter and a scroll pump (Scroll Meister). The inlet/sampling cell time lag, which varies with altitude, ranges between 2.5 s (25 kft) and 5 s (at the surface), and all final data have been appropriately time shifted to account for this.

In an effort to simplify and reduce costs, the pressure vessel (enclosure) was designed from a stock 12 inch OD 6061-T6
Aluminum pipe. The walls were turned down with a lathe in equidistant sections along its longitudinal axis, leaving thicker sections in the middle and end to accommodate the mounting of endplates and rack mounting points. The internal spectrometer assembly was suspended via vibration isolators to the outer shell, while the vessel was mounted to the rack with wire rope shock isolators. One endplate was used for easy access to the DFG module, while the opposing end was used
to feed through the gas and electrical connections and serve as a mounting plate for the air conditioner. However, during the first two field campaigns in the summer of 2016 and the winter 2017, the sealing area on the vessel end surfaces was poor resulting in a higher leak rate. This was subsequently rectified by increasing the sealing surface on both end plates, replacing an o-ring with a flat 10 mm wide rubber gasket. This significantly reduced enclosure pressure changes from values as high as 41 torr when the B200 cabin pressure changed by 144 torr as the aircraft ascended from 0.5-km to 9.1-km over the course of
12 minutes during the summer 2016 campaign to values as low as 0.3-torr change in the enclosure when the cabin pressure



changed by 19 torr during an ascent from 0.5-km to 4.5-km during the fall 2017 campaign. Sometimes rapid cabin pressure changes occurred about 1-minute or less prior to the aircraft changing altitude. Since such cabin pressure changes can occur over time periods much faster than can be captured by frequent zeroing and/or calibration, optical enclosure pressure stabilization is required for robust high performance. The poor pressure stabilization during the summer 2016 campaign

provided us with a direct quantitative figure of merit in terms of pressure change per unit time that must be achieved for high performance. During flight, enclosure pressures are maintained around 615 torr by pumping on the enclosure while employing an MKS 640A pressure controller and adding a small controlled flow (~ 0.2-slm) of zero air into the enclosure.

As shown in Fig. 1, the optical enclosure is thermally insulated while the temperature of the entire optical train is controlled.

The flow system computer controls all functionality (flows, pressures, temperatures) and sequencing of the flow system in accordance with commands from the spectrometer computer system. The system operates in three modes; Ambient, Background, and Calibration Standard. In the Ambient mode, outside ambient air is drawn through the multipass cell for 5-7 minutes after which the Background mode is engaged. Here ultra-pure air (Scott Marrin) from the air cylinder is fed to the inlet tip at a higher flow rate than the cell intake flow, thereby excluding ambient air from the system and allowing

instrument backgrounds to be recorded. In total, zero air is directed into the inlet for 90 seconds, which includes 30 seconds of background acquisitions and 30-second delay times before and after each background to allow sufficient time for flushing and system stabilization. After ~ 90 s, the Ambient mode is engaged again and the cycle starts over. During the Calibration Standard mode, a known mixing ratio $C_2H_6/CH_4$ (20/ 2000 ppbv) is fed into the zero air stream by a flow controller, which is then added to the inlet. Calibration standards are measured before and after each flight. During Ambient and Background

modes, a small suck-back flow (0.3-slm) is engaged to draw away any residual standard trapped in the addition line.

Acquisition of zero air backgrounds throughout each flight not only chemically zeros the entire gas handling flow path, which is important for elimination of outgassing effects after high transient concentration sampling, but also removes non-zero retrieved ethane mixing ratios due to optical effects. This frequent zeroing further allows us to assess instrument

precisions throughout each flight by fitting the zero air background spectra during each zeroing period. In comparison to previous systems the flow system here is made less complex by replacing heated scrubber systems and more complicated calibration systems with zero air and calibration gas cylinders as well as simplified gas handling paths. By controlling the air- and standard- gas flow rates, known concentrations can be generated and are used to verify the instrument accuracy and precision.

**2.5 Air Conditioning and Uninterruptible Power System**

The air conditioning system was designed to minimize size and complexity and consists of two (Qmax=341W each) thermoelectric cooler (TEC) elements attached to one endplate of the enclosure and adjacent fans attached to the inside





enclosure wall to circulate the air. The enclosure internal temperature was set to operate at 26 C +/- 0.5 C for a cabin temperature range of 10-30 C. Compared to previous system designs, this is less complex, but has a large power

consumption and degradation in cooling efficiency when the difference between enclosure and cabin temperatures reaches >10 C. The system warmup time is 60-90 minutes from a cold start depending on ambient temperature and is mainly dependent on the instrument thermal mass reaching operating temperature. Attached to the optical enclosure is thermal insulating foam (thermal conductivity 0.035 W/(m*K)). The UPS system keeps all system components running for a minimum of 30 minutes except the air-conditioning and vacuum pump during power switch overs from ground to aircraft

power and during refueling. However, depending upon the ambient temperature, not keeping enclosure temperature constant during no-power aircraft operations requires 20-30 minutes for the instrument to re-stabilize in some cases.

**2.6 Signal Processing and Software**

This section provides an overview of the various software processing modules, and more detailed information will be presented in subsequent sections. The computer software is based on object-oriented LabVIEW and uses standard and

custom plug-in software modules with the National Instruments Distributed Control and Automation Framework (DCAF) as well as custom FPGA code. To suppress asynchronous noise, the arbitrary waveform generator and data acquisition boards are controlled and phase locked by the FPGA. An 800 Hz sawtooth waveform with a smooth recovery function drives the laser scan current. After averaging the CD, AMD, and RD signals onboard the FPGA to the desired time resolution, these signals are sent to the host for processing in the following steps:


1. Common mode noise is mathematically removed for every measurement update by subtracting the AMD signal from the CD signal using the expression CD-R*AMD, where R is calculated as the ratio between the CD and AMD scan amplitudes. This provides a mathematical balancing of the CD and AMD powers.

2. The CD-R*AMD signal is wavelength locked by sub-channel shifting the recorded CD-R*AMD spectra using the $C_2H_6$ high concentration cell in the RD path as a reference. Here we utilize a manifold of at least 30 individual strong ethane vibrational-rotational lines spanning a 0.2 cm$^{-1}$ range centered at ~ 2996.8 cm$^{-1}$, which are the same as employed by Yacovitch et al. (2014). This simplified approach was found to be sufficient compared to the CAMS-1 instrument, where the laser current is used to actively stabilize the wavelength in a feedback loop.


3. Residual instrument noise is removed by periodically recording and subtracting the instrument spectral background by introducing ultra-pure air into the multi-pass cell for a period of 90 s, and repeated every 5 to 7 minutes before a new background is acquired, as previously discussed. This yields an ambient duty cycle ranging between 83-88%.



4. To retrieve the measured concentration, section 2.7 discusses further details of the fitting procedures employed in determining ambient ethane concentrations using background-subtracted spectra, and measurements of cell pressure, temperature, and path length along with spectroscopic parameters from the infrared database provided by Harrison et al. (2010) using the Beer-Lambert absorption law.

The software fits multiple absorption features in the same scan window as well as interference deconvolution. Figure 5 shows an example of fitting out multiple species ($C_2H_6$, $CH_4$) in the same scan, but in this case the $CH_4$ line strength only yields measurement precisions of ~30-40 ppbv, as the priority for scan window selection was aimed to obtain maximum precision of $C_2H_6$. As a DFG-based system has a wide and flexible tuning range and judicial selection of the wavelength region can accommodate multiple species to fit different measurement requirements. All measurement- and housekeeping-
data as well as unprocessed (raw) CD, AMD and RD spectra are stored on a local USB/SD drive that can be accessed via a router by either Wi-Fi or LAN connections.

A major effort was devoted to operating the instrument autonomously during flight, allowing the instrument, depending on the "instrument state", to take pre-programmed actions such as pause, restart, safe-state or shut down of individual
tasks/software modules or the complete instrument if needed. All actions and error messages are logged to help trace potential issues later. At power-up, the instrument runs a series of "checks" and if passed enters "ambient" measurement mode without the need to "calibrate" as the measurement is based on first-principle Beer-Lambert Absorption law.

**2.7 Direct Absorption Spectroscopy**

Figure 5 shows a simulation of the resulting absorption spectrum for 1 ppbv of ethane employing typical conditions of temperature (26.6 C), pressure (73.2 torr) and a path-length of 4760 cm using Voigt line profiles and the Harrison et al. (2010) database for ethane. As discussed by Yacovitch et al. (2014), the HITRAN database for ethane does not satisfactorily reproduce the ethane spectrum in the 3 μm region, and this was further verified by our comparisons of the direct absorption results with independently calibrated standards. We also show the spectrum for 2 ppmv of methane using the 2016 HITRAN
database. This simulation closely approximates the scan wavelength range used during the ACT-America studies. As can be seen, methane introduces a positive interference on ethane. At the ethane linecenter, these simulations indicate that 2 ppmv of methane produces an error of + 347 pptv on ethane. This is in close agreement with laboratory results employing calibrated methane standards. Such measurements reveal that 2 ppmv methane results in an error of + 342 pptv on the retrieved ethane. Although we can remove this interference using the methane line at 2997 cm$^{-1}$, we instead employ the more
precise onboard measurements of methane from a PICARRO G2401-m calibrated in flight with standards traceable to the WMO X2014 scale (ORNL dataset reference). Once the data has been carefully time shifted to match ambient features, this




is accomplished by subtracting the PICARRO methane values times the 342 pptv/2000 ppb factor from our retrieved initial ethane values. Future instrument configurations will employ the 2997 cm$^{-1}$ absorption line and stronger CH$_4$ features to remove this interference without the dependence on another instrument.


A key requirement for maintenance free operation is the implementation of absolute first principle direct absorption measurements via the Beer-Lambert Absorption Law:

$$\frac{I}{I_o} = e^{-\sigma\,PL\,MR\,N} = e^{-A} \quad (1)$$

Here, I and Io are the transmitted and incident intensities, respectively, acquired from measurements of the CD at each
wavelength step, σ is the absorption cross section, PL is the absorption path-length, MR is the mixing ratio of ethane, N is the air number density flowing in the absorption path at the sampling temperature and pressure, and A is the resulting absorbance (base e). Prior to each ambient acquisition, background spectra (CD-R*AMD)$_{Bkg}$ are acquired by directing zero air into the inlet. The background spectra are averaged and used to subtract from the subsequent ambient acquisitions to obtain a relatively flat transmission spectrum. The remaining baseline curvature is removed using a 3-5 order polynomial
function. Io values are determined on the CD signal at each wavelength step using a low-pass filter to remove the absorption feature. The high-pass filtering of (CD-R*AMD) provide measurements of the differential absorption spectrum as dI values at each wavelength step. We then calculate an absorbance at each step from:

$$A = -ln\frac{Io-dI}{Io} = \sigma\,PL\,MR\,N \quad (2)$$

Employing the Marquardt-Levenberg non-linear fitting algorithm (Marquardt, 1963; Levenberg, 1944) with these
absorbance measurements along with spectral parameters from Harrison et al. (2010) for ethane and from the 2016 HITRAN database for methane and water, Voigt line shapes, and measurements of pressure, temperature, and path-length, we calculate the best-fit absorbance profile. The system software supports fitting the integrated area of each spectral feature employing the appropriate integrated absorption cross sections in determining mixing ratios via Eq. (2). However, it was found that fitting to a peak absorbance using the line-center absorption cross section was less susceptible to baseline noise, particularly
from optical fringes, and slow changes in baseline curvature. This latter approach was therefore used throughout the campaigns in retrieving ethane mixing ratios. In this approach, a peak ethane absorption cross section was determined at line center σ(ν0) of 9.03 x10$^{-18}$ cm$^2$ molecule$^{-1}$ from the Voigt simulations shown in Fig. 5 employing the Harrison et al. (2010) database for the 2996.85 cm$^{-1}$ manifold of ethane lines after accounting for the relatively broad absorption baseline pedestal shown. Figure 6 shows raw and fitted spectra acquired for ethane (4.23 ppbv) and methane in our laboratory. The methane
feature underlying the ethane is still present, but is not visible here since the methane almost perfectly overlaps with ethane. Since the width of this fit is not optimized for the methane feature on the right, its absorbance is underestimated.





The overall cross section uncertainty quoted by Harrison et al. (2010) is ± 4%, and since that study determines the results at various pressures, including those around 76 torr, we assume that the uncertainty in the exact individual line positions are
taken into account. This is important since our recorded ethane feature is the convolution of at least 30 individual lines at our 73 torr sampling pressure, and small line position errors could add to the uncertainty of our deduced peak linecenter absorption cross section above.

To further validate the implementation of the direct absorption approach as well as the linecenter absorption cross section
given above, we introduce known $C_2H_6/CH_4$ calibration standards in compressed gas cylinders from Scott Marrin into the inlet before and after each flight. CAMS direct absorption measurements retrieved ethane mixing ratios that were too low by 6% and all ACT-America data have been subsequently raised by this number. However, laboratory measurements following the spring 2018 4th deployment phase, we carried out extensive standards inter-comparisons with other ethane standards from this same manufacturer as well as two standards (mixing ratios around 3 and 0.3 ppbv) from the NOAA ESRL Global
Monitoring Division. These two standards, which were prepared gravimetrically by the NOAA/ESRL Global Monitoring Division and used in programmable flask package (PFP) ethane measurements discussed by Baier et al. (2019), were further measured by Detlev Helmig's Atmospheric Research Laboratory at the University of Colorado Institute of Arctic and Alpine Research using standards tied to the Global Greenhouse Gas Reference Network (see for example Helmig et al., 2016). Measurements of the latter two ethane standards indicated that our assigned ethane values employed during ACT-America
were on average too high by 4.5%, and hence our original direct absorption measurements produced ethane values closer to these globally accepted values. We note that the comparisons of our ambient ethane data that will be shown relative to the NOAA (PFP) measurements acquired on the same B200 aircraft during ACT-America have not been adjusted by this 4.5% factor but are based upon our 6% over-corrected submitted data to the NASA archive. Nevertheless, the retrieved ethane mixing ratios, particularly during the 4th spring 2018 deployment, are accurate on average to within ± 6% range. Figure 9 (to
be discussed in section 3.3) shows comparisons of airborne CAMS and PFP ethane data.

## 2.8 Background Acquisitions, Diagnostics, & Data Handling

This section will discuss additional sources of uncertainty associated with background changes measured during zero air background additions. Backgrounds are acquired by overflowing the inlet with zero air every 5 to 7 minutes using ultrapure compressed air. Initially, a 5-minute ambient period was chosen before a new background was acquired. Numerous
instrument improvements during each field deployment phase enabled zero air background subtraction to be extended to 7-minutes by the 4th campaign in the spring of 2018. These backgrounds not only provide a semi-continuous assessment of the instrument measurement precision, but also represent an important component of the measurement accuracy. In contrast to many other studies that report a single precision and accuracy assessment for an entire study, we report the measurement performance with every ambient cycle, and over the course of each mission these assessments cover the full range of aircraft





maneuvers of pitch, roll, aircraft ascents and descents, cabin pressure and temperature changes as well as vibrations. The resulting data, reported as histograms, thus provide a more representative picture of the true instrument performance.

Figure 7a shows two background ($Bkg_n$, $Bkg_{n+1}$) acquisition periods during the 4th deployment phase during the spring of 2018. The first background (Period A) is prior to the ambient period and the second one is after the ambient period (Period

B). The entire 90-second background periods, which includes 30-seconds for background acquisition plus the two flushing periods are shown. Each of the ambient fit results shown here employed $Bkg_n$ acquired during Period A to subtract, and thus as a means to remove the optical background structure. The backgrounds are fit and treated with an identical procedure as the ambient acquisitions, that are using the previous background to subtract and remove residual optical noise. In the case of Period A, we show the residual fit of $Bkg_n$ acquired during this period minus $Bkg_{n-1}$, acquired 7 minutes prior (not shown).

As illustrated, the fit of the resulting background difference ($Bkg_n$ - $Bkg_{n-1}$) yields a stable background difference (0.020 ± 0.018 ppb) close to zero. The instrument precision (or limit of detection, LOD) was determined for each associated ambient period from the standard deviation of this background fit difference. We assign a single LOD based on this precision for each ambient period, and these are plotted as error bars with each 1-second ambient result. This standard deviation is close to that determined for the ambient acquisition period indicated (± 0.031 ppb) and further supports our LOD estimates. As true

ambient variability cannot be ruled out, the larger ambient imprecision compared to the background fit difference is not surprising. During the last 3-seconds of each background cycle before valve switching back to ambient, we plot a 3-second Snapshot of ultra-pure air (background), where we subtract the background acquired during the past 30-seconds from this 3-second Snapshot as a means to highlight fast changes in the present background. This background difference is annotated in red. Under perfectly stable conditions, one should expect values around 0 without significant background drifts over the

course of the 30 seconds. As discussed previously, the new background ($Bkg_n$ in this case acquired over the full 30 seconds) is then used for the subsequent ambient spectra. We emphasize here that the 3-second Snapshot period is a diagnostic meant to show fast background changes relative to the full 30-second new backgrounds that are employed for the next ambient period. During Period B, we show a plot of the newest background minus the previous background ($Bkg_{n+1}$ - $Bkg_n$). This results in a value of 0.051 ± 0.035 ppb, and a Snapshot value of -0.014 ± 0.029 ppb.


During stable instrument performance, as indicated in Fig. 7a, not only should the 3-second Snapshot values lie around zero, within the indicated precision, but also the background differences before and after each ambient should be equivalent within the measurement precisions. In this case the background during Period A ($Bkg_n$-$Bkg_{n-1}$ = 0.020 ± 0.018 ppb) is equivalent to the background during Period B ($Bkg_{n+1}$-$Bkg_n$ = 0.051 ± 0.035 ppb).


However, perturbations from the various sources mentioned above during flight will show offsets not only between adjacent background differences, but also between the latest background and the 3-second Snapshot values. Figure 7b exhibits such



extreme behavior during the 1st field deployment phase before the enclosure was pressure sealed. Here we highlight the pre-ambient background fit difference in the shaded region when the pressure in the enclosure changed by 4.4 torr. As shown,

this resulted in a large offset of 0.730-ppbv ± 0.071-ppbv. In addition, the Snapshot value of 0.207 ± 0.007-ppbv during the 3-second period (shown by the notch) not only yields a value far removed from zero, resulting from the drift, but also a background change of 0.523-ppb. The 2nd background period, furthermore, revealed an additional background change from the 0.207-ppb value to 0.113-ppb. As a result, our ethane data from the 1st deployment phase conducted during the summer of 2016 has to take into account these performance limitations.


Figure 7c shows the background behavior during the 3rd deployment phase in the fall of 2017 that is typically observed once the temperature of the system has been re-stabilized for at least 30-minutes after take and after the enclosure pressurization was fixed. As can be seen, large fluctuations in cabin pressure no longer affect the enclosure pressure, and hence the background structure.


Although the background profiles, and hence the quality of the ambient ethane data, were significantly improved during the 4th field deployment phase, as shown in Fig. 7a, we still observed moderate background shifts even after system temperature stabilization. Figure 7d, which was acquired on the same day as Fig. 7a, provides one such example. The background data during Period A reveals essentially the same excellent performance as Fig. 7a. However, the background data in Period B

reveals a residual system sensitivity to what we believe are caused by rapid changes in aircraft pitch as the aircraft was preparing for landing, but have been observed during other occasions even after improved optics stabilization. Although the precisions are still excellent, here the background jumps from an average value of -0.002 ppbv to 0.188-ppbv, and the 3-second Snapshot period previously unobservable becomes immediately evident during the 2nd change in aircraft pitch in Period B. To account for such additional time background changes we applied an additional correction to the final ambient

ethane data. Referring to Fig. 7d, we linearly interpolate the background data between the 3-second Snapshot of Period A (which represents the new background that is applied to the subsequent ambient data) and the average background data at the start of Period B. This linear background temporal interpolation, which is subtracted from the ambient data between the two background periods, accounts for linear background drifts. Obviously, non-linear drifts or jumps in the true background will cause data errors. However, this would show up as artificial ambient ethane structure that is uncorrelated with other

measurements. Our subsequent data analysis using our ethane data flags such time periods, especially where there are large background changes and/or the ethane data shows such an artificial time dependence. Using this same logic for the next ambient period, we interpolate between the 0.035 ppb Snapshot in Period B to the mean background at the start of the next background period (0.038 ± 0.032 ppb). We estimate the component of uncertainty due to such background changes over each ambient time period by 1/2 of the difference of the background values at the two end points (the value during the



Snapshot period and the mean value at the beginning of the next background period. Section 3.1 further discusses the various components to our estimated total measurement uncertainties.

## 3 System Improvements During Each ACT-America Airborne field deployment phase & Comparisons of CAMS-2 with CAMS-1

CAMS-2 was designed to realize a small, lightweight, fully autonomous, and calibration-free instrument. While several
aspects of the CAMS-1 design were inherited by CAMS-2, a number of new approaches were implemented for CAMS-2 and continually evolved for the entire duration of the field deployments. Improvements and simplifications between the CAMS-2 and CAMS-1 designs will be compared at the end of this section.

### 3.1 Improvements with each Field Deployment Phase

The most impactful improvements were implemented before the 3rd and 4th mission field deployment phases. Prior to the
3rd field deployment phase, the enclosure pressure was stabilized and in addition, the optimum PPLN phase matching temperature was de-tuned to reduce a small halo spatial emission mode exiting the PPLN crystal. Although this substantially reduced mid-IR power, it significantly improved the matching between the CD and AMD arms, resulting in improved performance, as will be further discussed below in this section. Prior to the 4th field deployment phase in the spring of 2018, we further addressed the mechanical stability of various optical components.


The mechanical construction between the MP cell and the launching optics were improved by stabilizing bars reducing movements induced by accelerations. An improvement of ~2x in baseline concentration stability was verified by inducing instrument tip-and-tilt actions in the lab. The mechanical stability is not only affected by accelerations, but also by enclosure temperature variations that reached ± 0.5 C over 5 min during normal flight operations. Two potential causes were identified:
1) lack of efficient air exchange between the thermoelectric cooling unit and optical components; and 2) low thermoelectric efficiency at > 30 C cabin temperatures. This is in contrast to CAMS-1 temperature control of ± 0.1 C and is a contributor to performance degradation in non-laboratory environmental conditions. This furthermore necessitated at least 30 minutes of re-stabilization before optimal performance was achieved after takeoff. Even though the temperature changes are relatively slow, they can affect the mechanical stability through thermal expansion, but also alter the optical properties of active and
passive fibers, as well as perturb the non-linear optical frequency generation process in the PPLN crystal. These components have previously been determined to be sensitive to temperature variations affecting short and long term drifts in the spectroscopic baseline (Weibring et al. 2006, 2007). Therefore the PPLN compartment was insulated and the fiber trays were padded by foam similar to CAMS-1/DFGAS to insulate and slow down temperature variations as well as dampen fiber vibrations during airborne operation. This significantly reduced high frequency noise.






Small PPLN temperature instabilities can result in secondary effects that alter the temporal and spatial beam propagation of the highly focused beams through the PPLN. According to Zhi-Yuan Zhou et al. (2014), non-collinear focusing of the two beams into the PPLN crystal results in spatial deviation of the PPLN output from a gaussian beam and adding a donut shaped halo component to the output beam. The spatial evolution for propagation of such a beam deviates between the near and far

field compared to the ideal gaussian beam making it especially difficult to record and remove common mode noise with our subtraction approach, which is dependent on the fact that the spatial beam properties between CD and AMD detectors are only influenced by the cell transmission and not a spatial mismatch between the CD and AMD detector areas. This was confirmed to be present in CAMS-2 using a mid-infrared camera, while CAMS-1 did not show the same behavior. CAMS-1 and CAMS-2 use different focusing geometries resulting in a more non-collinear phase matching in the latter case.

Adjustments within the CAMS-2 design to attain minimal non-collinear phase matching could not be achieved. Therefore, CAMS-2 required temperature de-tuning of the PPLN crystal from the optimal power generating phase-matching temperature to suppress residual non-collinear phase matching, improving the spatial beam shape and degree of matching in the CD and AMD arms. The drawback of operating on the edge of the phase matching bandwidth resulted in larger power instability induced by small ambient temperature perturbations.


The resulting low DFG power of 10-20 µW placed stringent requirements on the electrical noise of the system to ensure that the system is operating in the shot noise regime. New low noise switching power supplies and filtering dedicated for the detector pre-amplifiers and sequential noise filtering were applied (See section 2.3). A low noise MCT detector type with a high shunt resistance was selected. The small detector surface area was mitigated by an attached half-ball immersion lens making the apparent detector area 1 mm diameter, allowing less demanding focusing conditions. The drawback using a

detector with a ball lens is that regardless of the incoming beam angle there will always be a reflection going straight back into the incoming beam (cateye effect), potentially causing optical feedback and associated fringing. While DFG is immune to direct feedback, mid-IR laser sources such as Quantum Cascade Laser are known to be affected by optical feedback. However, reflected/scattered beams in the optical system still can produce optical fringe noise between enclosure walls or

optical components. To minimize such effects, optical components were tilted and reflected beams removed by beam stops.

The various performance-inhibiting thermal issues were mitigated before the 4th field deployment phase and resulted in robust high performance during airborne operations. The results can immediately be seen using histograms by comparing the LODs determined from the background measurements during the 3rd field deployment phase in the fall of 2017 in Fig. 8a

with those in Fig. 8b acquired during the 4th field deployment phase in the spring of 2018.

We show in these figures histograms of the resulting 1-second background precisions acquired with each ambient acquisition period for the entire 3rd and 4th field deployment phases and the resulting log normal fits and associated mode of the fits



reveal both instantaneous noise as well as instrument drift since last background recording. In Fig. 8a, even though the mode

of the log normal fit reveals an excellent LOD of 0.040-ppbv during the 3rd field deployment phase, one can immediately observe a second distribution mode by the rather large tail in the histograms out to values as high as 0.340-ppbv. By comparison, the two histograms shown in Fig. 8b (one in the PBL with pressure altitudes < 2km and one in calmer, less turbulent free troposphere air at altitudes > 2km) both reveal only single mode distributions with log normal mode fit values of 0.040-ppbv throughout the entire 4th campaign, and hence more stable instrument performance. One also observes a

considerable number of background values in the 0.030 to 0.040-ppbv range. It is important to note that these comparisons, which reveal the effectiveness of our improvements, were acquired over all the various aircraft perturbations during both field deployment phases (i.e., pitch, roll, yaw, cabin pressure and temperature changes, and vibrations) which provided challenges to our ethane measurements. By showing LOD histograms under all such conditions, particularly in both the turbulent boundary layer and the calmer free troposphere, furthermore, accentuates the dynamic nature of airborne

performance and reinforces the fact that a single performance estimate does not truly capture this dynamic performance.

As previously stated, the measurement LODs only reveal part of the performance story as changes in background structure acquired during zeroing between ambient acquisitions dictates the overall total measurement uncertainty (TMU). The TMU at the 1-$\sigma$ level is comprised of 5-terms, and these terms were added in quadrature. These terms are: 1) the LOD based upon

the background precisions prior to each ambient acquisition period; 2) temporal changes in the background differences over the course of each ambient acquisition, as discussed in the previous section; 3) the uncertainty in the methane interference correction (0.342-ppbv/2000 ppbv $CH_4 \pm 0.006$), as determined in the laboratory; 4) the PICARRO methane measurement error ($\pm$ 1-ppbv, https://doi.org/10.3334/ORNLDAAC/1556); and 5) the uncertainty in the fitting correction factor employing the input calibration standards. Table 1, shows the estimated TMU for 3 of the field deployment phases estimated during all

the ambient measurements for ethane values > 0.5-ppbv. The TMUs are given both in absolute ethane mixing ratios as well as the percent of the ambient values. The temporal changes in the background differences comprises the largest contribution to these TMUs. As can be seen, the median LODs are all less than 54% of the TMU values, which for the latter values fall within the 0.095 to 0.164-ppbv range (4% to ~ 10% of the ambient values).

Although the TMUs are quite good, they can be improved significantly by further mechanical stabilization of the optics towards aircraft pitch changes as well as improved temperature control of the optics within the enclosure. As with our dynamic LOD estimates, these dynamic TMU estimates truly reveal the instrument performance range over the full range of aircraft maneuvers. The comparisons of the CAMS ethane measurements with the flask package (PFP) measurements acquired by NOAA on the same aircraft (Baier et al., 2019), discussed in Section 3.3, further supports the lower range of our

TMU estimates. In 2 of the 3 field deployment phases the median TMU as a % of ambient values falls in the 4 to 5.2% range, which as will be seen in Section 3.3 yields nearly identical CAMS ethane values as the PFP values to within 4.5 ±





0.8%. This also includes the third field deployment phase where TMU estimate yields a median value of 9.9%. Perhaps our TMU estimates in some cases over-accentuates the background drift component of uncertainty.

## 3.2 Comparison of CAMS-2 to CAMS-1 Performance

In this section we compare the CAMS-2 ethane airborne performance from the 4th field deployment phase with those from airborne CAMS-1 ethane measurements employing a much heavier, larger, and more complex airborne system. CAMS-1 employs 2-f harmonic detection, while CAMS-2 employs direct absorption spectroscopy. Direct absorption spectroscopy has the advantage to yield an absolute measurement based upon fundamental principles. For comparison, CAMS-1 employed second harmonic detection, which reduces the signal strength by ~ 2-3 times and the precision by the square root of (2-3). If,
however, the measured signal contains large technical (electronic, optical, environmental) noise that is uncorrelated with the spectral scan, phase sensitive second harmonic detection effectively can suppress such noise to a greater extent than direct absorption techniques. However, with judicial design and attention to details, technical noise can be suppressed and minimized. We compare the 0.030 - 0.040-ppbv 1-second CAMS-2 airborne ethane performance here to the median 1-second value of 0.025-ppbv for ethane in CAMS-1. As different absorption features, sampling pressures, and path lengths
were used, this comparison requires one to translate the above concentrations to minimal detectable line-center absorbance values, $A_{min}$. CAMS-1 operates on a manifold of ethane lines at 2986 cm$^{-1}$ compared with the 2996 cm$^{-1}$ manifold for CAMS-2. CAMS-1 operates at a sampling pressure of 50 torr using a 89.6 m pathlength, while CAMS-2 operates at 73 torr using a 47.6 m pathlength. Using a mid-range CAMS-2 value of 0.035 ppbv compared to 0.025 ppbv in CAMS-1, we calculate a ratio $(A_{min})$CAMS-2/ $(A_{min})$CAMS-1 = 3.5 x 10$^{-6}$/2.8 x10$^{-6}$ = 1.3. That is, we achieve nearly comparable airborne
ethane performance in an instrument that is over a factor of 3 times lighter (when one folds in no operator and seat), significantly less complicated to operate and approximately half the size. Our 0.015 - 0.020 ppbv precisions during laboratory conditions further suggests an even more favorable CAMS-2 comparison if one further addresses the remaining temperature stability issues and the residual sensitivity to rapid pitch maneuvers.

## 3.3 Airborne Comparisons of CAMS-2 Ethane Measurements with NOAA's Programmable Flask Package System

As discussed in Section 2.7, ethane standards comparisons with the NOAA/ESRL's Global Monitoring Division as well as ethane standards tied to the Global VOC program vis-à-vis comparisons with standards from the INSTAAR Atmospheric Research Laboratory of D. Helmig lab revealed that our submitted ethane concentrations are too high by 4.5%. During each field deployment (II-IV), we carried out comparisons of our continuous 1-second ethane measurements (not corrected by the calibration standards comparisons) with the NOAA PFP results by averaging our results over the flask fill start and stop
times of the PFP system. This procedure is accurate during constant ethane mixing ratios when rapid ethane changes in plumes are not sampled. When sampling plumes, by contrast, one would need to know the exact temporal filling profile of the PFP system in order to modify the CAMS averaging kernel. This is further discussed by Baier et al. (2019). In plumes without taking this into account, one can thus obtain fast averages that are both too high, too low, and in agreement with the



PFP measurements, depending upon the slope of the changes. Thus, to reduce such effects, we exclude CAMS data whose
standard deviation over the PFP sampling period is greater than 0.5-ppbv. Excluding such data, linear regressions
(orthogonal distance regressions, ODR) of the fast CAMS ethane data averaged over the PFP time base (Y-axis) vs the PFP
measurements (X-axis) yields the results shown in Table 2. The average slope value for the 3 field deployment phases is
1.045, which exactly matches the calibration standards comparisons after the 4th field deployment phase. At present, we do
not have an explanation for the small but persistent negative intercepts that average to a value of -0.135 ppbv. This could
imply that either the PFP measurements could have a small positive interference or the CAMS direct absorption
measurements could have a small negative interference from the tails of nearby absorptions. These results suggest that our
Harrison et al. (2010) cross-section needs to be reduced by 4.5%. Figure 9 shows the regression plot, in which one point with
a standard deviation exceeding 0.5-ppbv was eliminated from the fit.

**4 Employing CAMS Ethane Measurements in Assessing Methane Sources**

As stated in the introduction, methane and ethane have common sources from oil and natural gas exploration and
production, coal mines, and wildfires. Ratios of ethane and methane measurements can be used to distinguish from biogenic
sources of methane. As one example, our CAMS-2 ethane results were employed by Barkley et al. (2019a) to estimate
methane emissions using a top down approach from coal and natural gas production in southwestern Pennsylvania. This
research concludes that while Environmental Protection Agency inventories appear to report emissions from coal accurately,
emissions from unconventional natural gas are underreported in the region by a factor of 2 to 8. In another example, in
Barkley et al. (2019b), ethane/methane slopes from large plumes across frontal boundaries in the Midwest are used to
differentiate between oil & gas and animal agriculture sources. In this case, the high ethane/methane ratios led to the
conclusion that oil & gas sources were responsible for a majority of the unaccounted methane emissions observed in the
frontal flights.


In another application, we show simultaneous ethane and methane measurements over the South-central United States to
derive ethane/methane slopes over various shale basins. Figure 10 shows one such example from the Southern deployment of
the Fall 2017 field campaign. Here we show simultaneous enhancements in ethane and methane from the many oil and
natural gas exploration and production activities over this region. These include the Permian, Eagle Ford, Barnett, and
Haynesville shale regions shown in Fig. 11. This figure employs fast measurements provided by CAMS-2 to derive
ethane/methane slopes shown on the flight tracks as colored points. This plot also shows the gridded methane emissions
from the 2012 EPA inventory (Maasakkers et al. 2012) as well as the wind directions and speeds. As can be seen, the
ethane/methane slopes over this region are highly variable and range from 0.5 to 29.1%. Fig. 11 highlights 4 individual
plumes by the black circles surrounding each plume. Figure 12a further shows the ethane and methane time series
corresponding to Plume 1, while Fig. 12b shows this for Plumes 2 & 3. Here the ethane/methane slopes, which range from

7.1% to 18.7%, reflect emissions primarily from the Barnett Shale and Eagle Ford Shale regions based upon proximity and wind direction. Although we have not yet carried out the same careful shale basin analysis as Peischl et al. (2018) for this region, the ethane/methane slopes of Fig. 11 fall in the same range (8.5% to 20.5%) as two study days reported by Peischl et al. (2018) as well as the 9.6% reported by Smith et al. (2015).

## 5 Summary

We present in this study a new autonomous airborne ethane instrument for fast 1-second measurements on the NASA B200 aircraft for ACT-America Studies based upon the CAMS-2 DFG spectrometer. This instrument is significantly smaller and lighter weight than its CAMS-1 predecessor and yields nearly comparable performance within a factor ~ 1.3. By operating autonomously, we eliminate the weight of ~250 - 300 pounds typically reserved for an operator and seat. The CAMS-2
instrument employs a pressure-stabilized and thermally controlled enclosure to avoid performance degradation due to aircraft cabin pressure and temperature changes.

This system reliably acquired high precision and fast ethane measurements on the B200 aircraft over several hundred flight hours during the five ACT-America field campaigns. The airborne performance was significantly improved with each
successive field deployment phase study, and we summarized herein the major upgraded design features to achieve these improvements. During the 4th field campaign, in the spring of 2018, we achieved 1-second ($1\sigma$) airborne ethane precisions reproducibly in the 30 - 40 parts-per-trillion (pptv) range in both the boundary layer and the less turbulent, free troposphere. To our knowledge, this performance is among some of the best reported to date for fast airborne ethane measurements. In both the laboratory and at times during steady airborne operation these precisions were as low as 15 - 20 pptv. Comparisons
with an onboard PFP ethane instrument results produced agreement to within $4.5 \pm 0.8\%$, values that are roughly consistent in 2 of the 3 cases with estimates of TMU based upon an error analysis. It is important to note that our LOD estimates and TMU estimates were dynamically determined over the full range of aircraft maneuvers and are thus more representative of instrument performance than a single estimate at a few select conditions.

**Data availability**

Measurement data is available on https://daac.ornl.gov/cgi-bin/dataset_lister.pl?p=37.



**Author contributions**

PW, DR, JW, AF – Developed, tested, and operated the new instrument during the ACT-America field campaigns. JD, HH, YC – Provided the methane measurements discussed in Section 4. BB, CS, BM - Provided NOAA's Portable Flask Package

(PFP) Ethane Measurements for airborne comparisons discussed in Section 3.3. KD, ZB and MO – In addition to providing critical support for the development activities discussed, but also helped in the analysis of ethane-methane slopes discussed in Section 4.

**Competing interests**

The authors declare, to their knowledge, no conflicts of interest with the submittal of this manuscript.

**Acknowledgement**

ACT-America is a NASA Earth Venture Suborbital 2 project funded by NASA's Earth Science Division. The authors acknowledge support from this division for Grant NNX15AW47G to the University of Colorado. We thank the ACT-America scientific and administrative team members, and especially the NASA Langley B200 aircraft crew, for their valuable support during the ACT-America field missions.

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





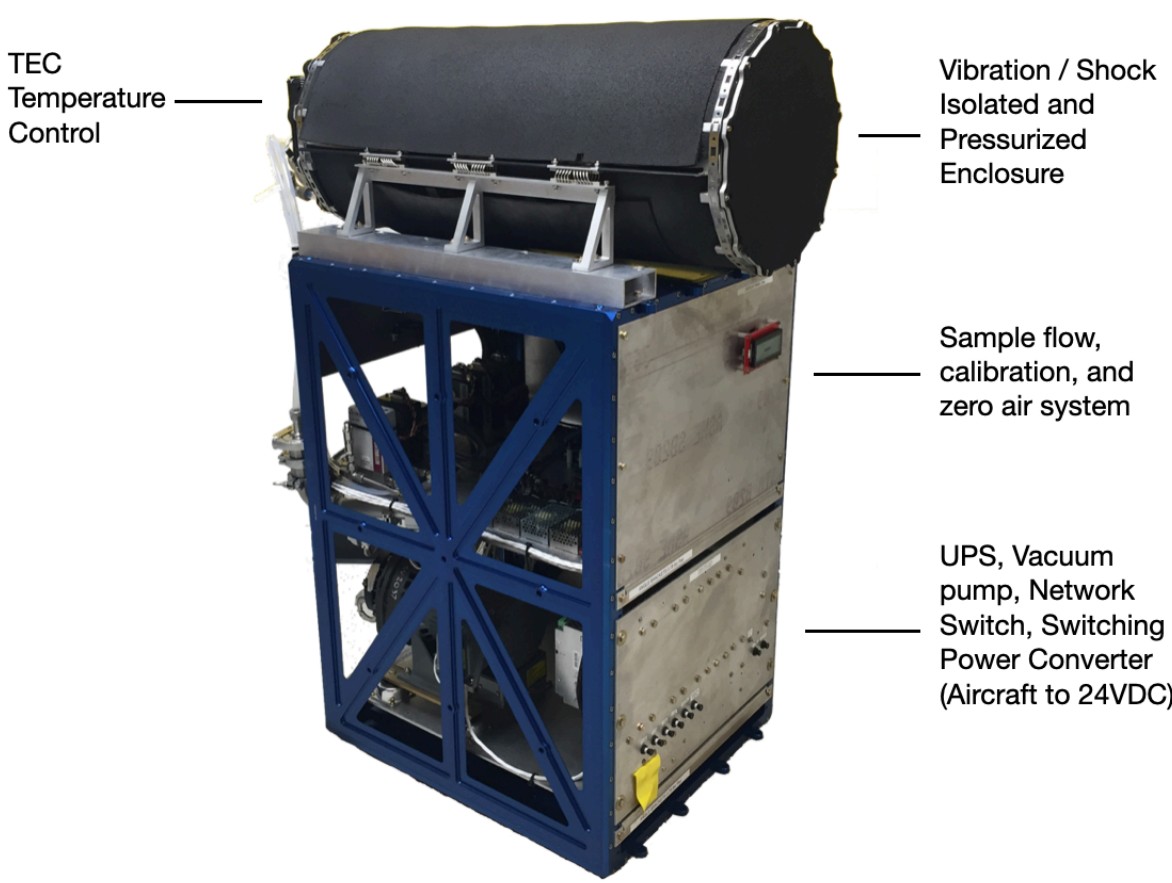

**Figure 1: Instrument Layout and Components.**





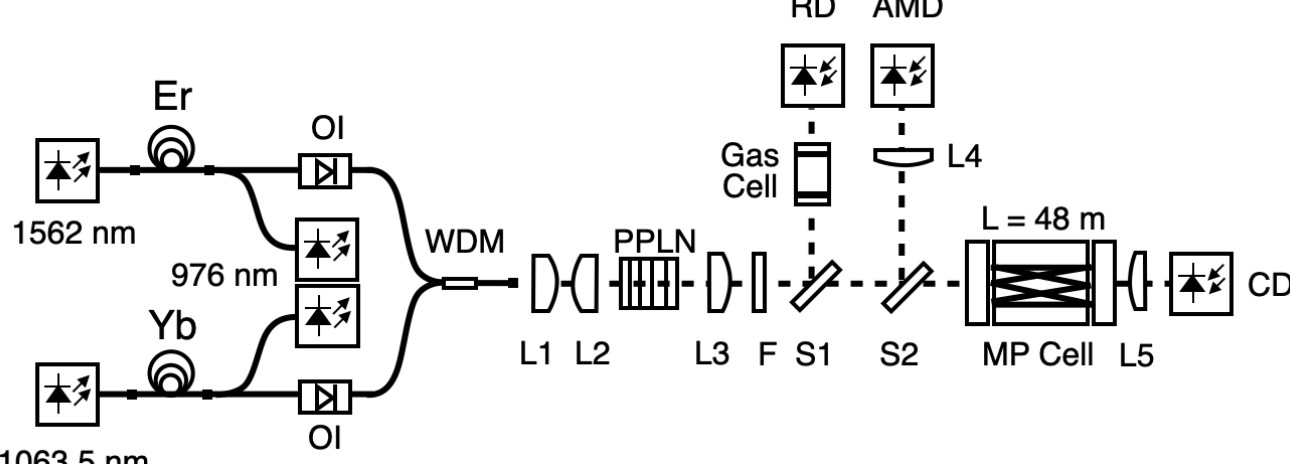

Mid-IR Power: <0.1 mW operating power
(0.8 mW max), 300 $\mu$W W$^2$ cm$^{-1}$

**Figure 2: Mid-IR Source Schematic. Er, Erbium doped fiber; Yb, Ytterbium doped fiber; OI, Optical Isolator; WDM, Wavelength**
**Division Multiplexer; L1-5, Lens; PPLN, Periodically-poled Lithium Niobate; F, Ge-Filter; S1-2, Beam-splitter; MP Cell, Multipass Cell; RD, Reference Detector; AMD, Amplitude Modulation Detector; CD, Cell detector.**

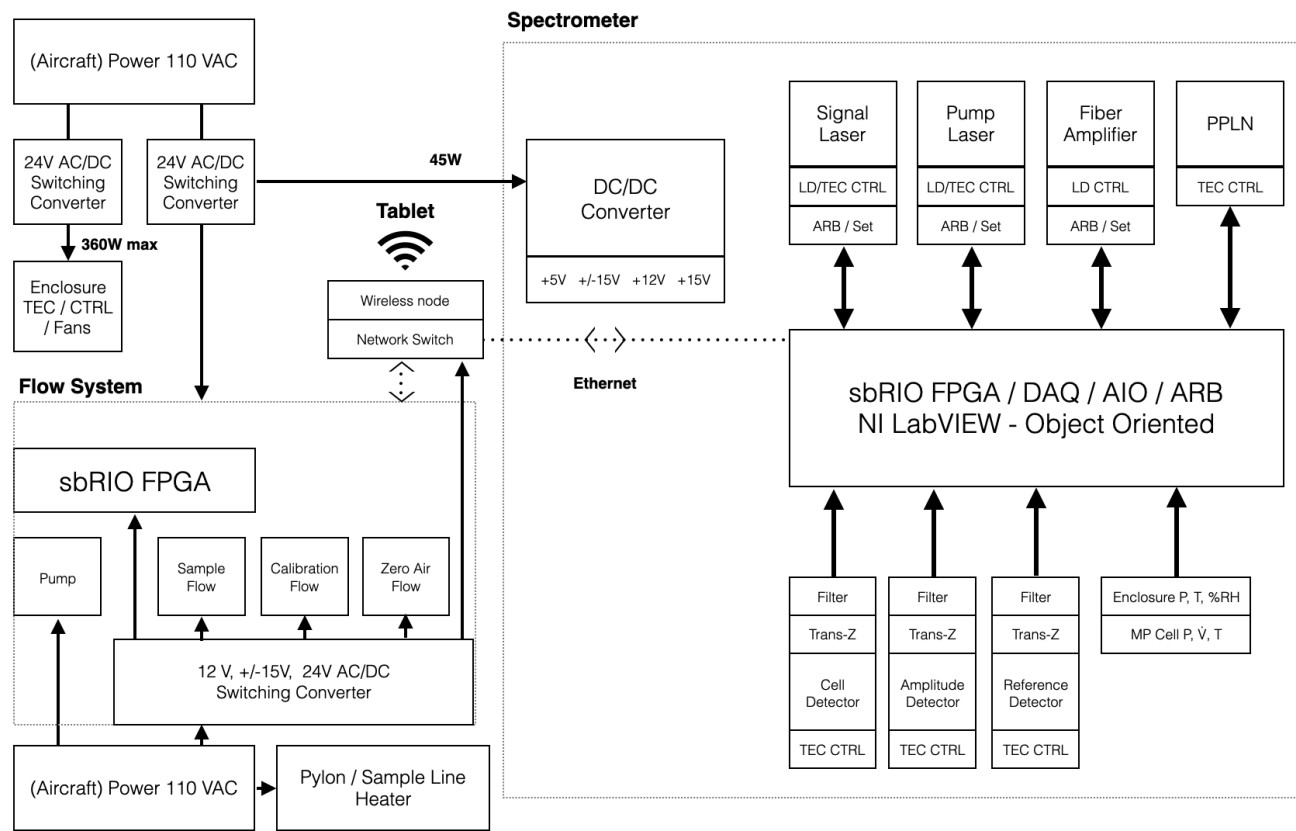

**Figure 3: Electronics and Power Schematic. LD, Laser Diode; TEC, Thermo-Electric Cooler; CTRL, Controller; ARB, Arbitrary**
**waveform output; Set, Set-point voltage output; DAQ, Data Acquisition; AIO, Analog Input-Output; Trans-Z, Transimpedance Amplifier; P, Pressure, T, Temperature; V, Volumetric Flow.**



**Figure 4: Flow System Diagram. V1-5, Pressure Relieve Valve; PC, Pressure Controller; FC, Flow Controller; FM, Flow Meter; MPC, Multi-pass Cell; CAL, Calibration Gas Cylinder; TEC AC, Thermo-Electric Air Conditioner; Filter, 3 μm Particle Filter.**




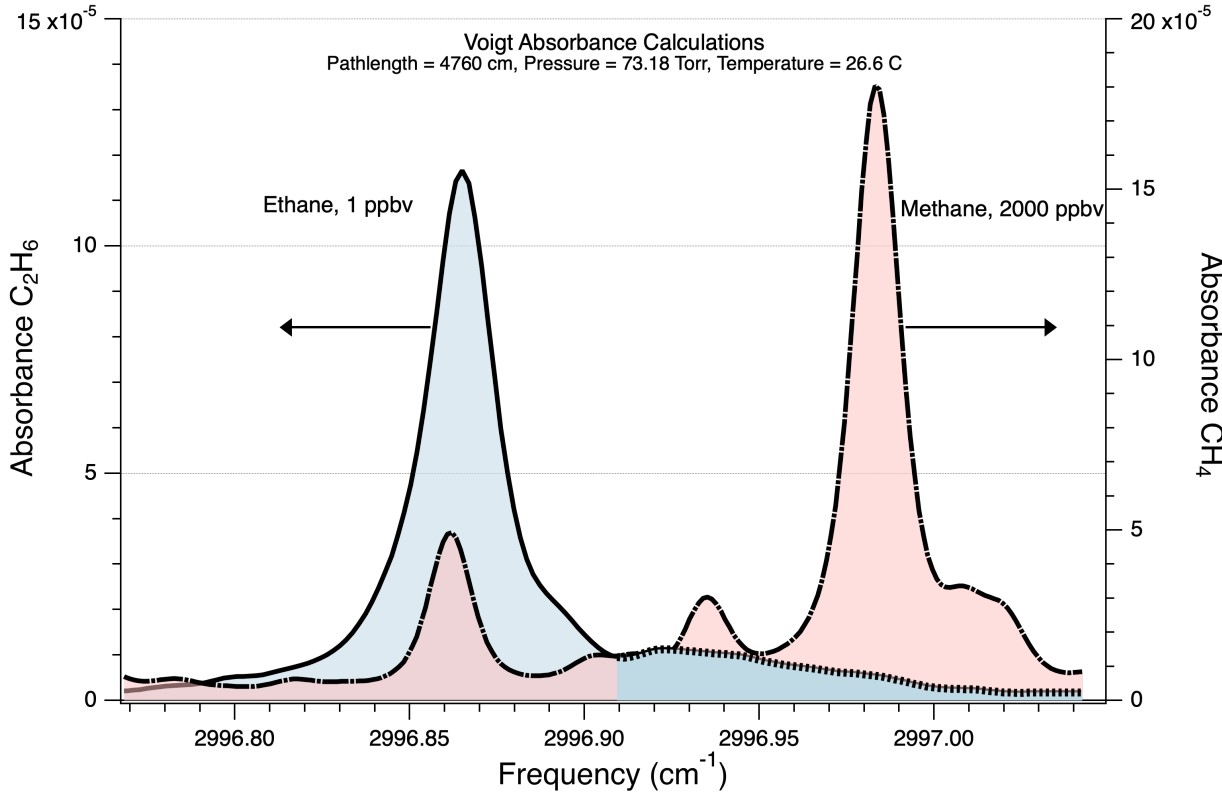

**Figure 5: Ethane & Methane Simulated Lines Using Harrison et al. [2010] and HITRAN 2016 Line Parameters. Note the rather large wavelength spread of the ethane background data caused by a multitude of small ethane and methane lines in the wings. In the text we refer to this as a pedestal and is further highlighted by the blue area at the high frequency side of the ethane absorption wing.**






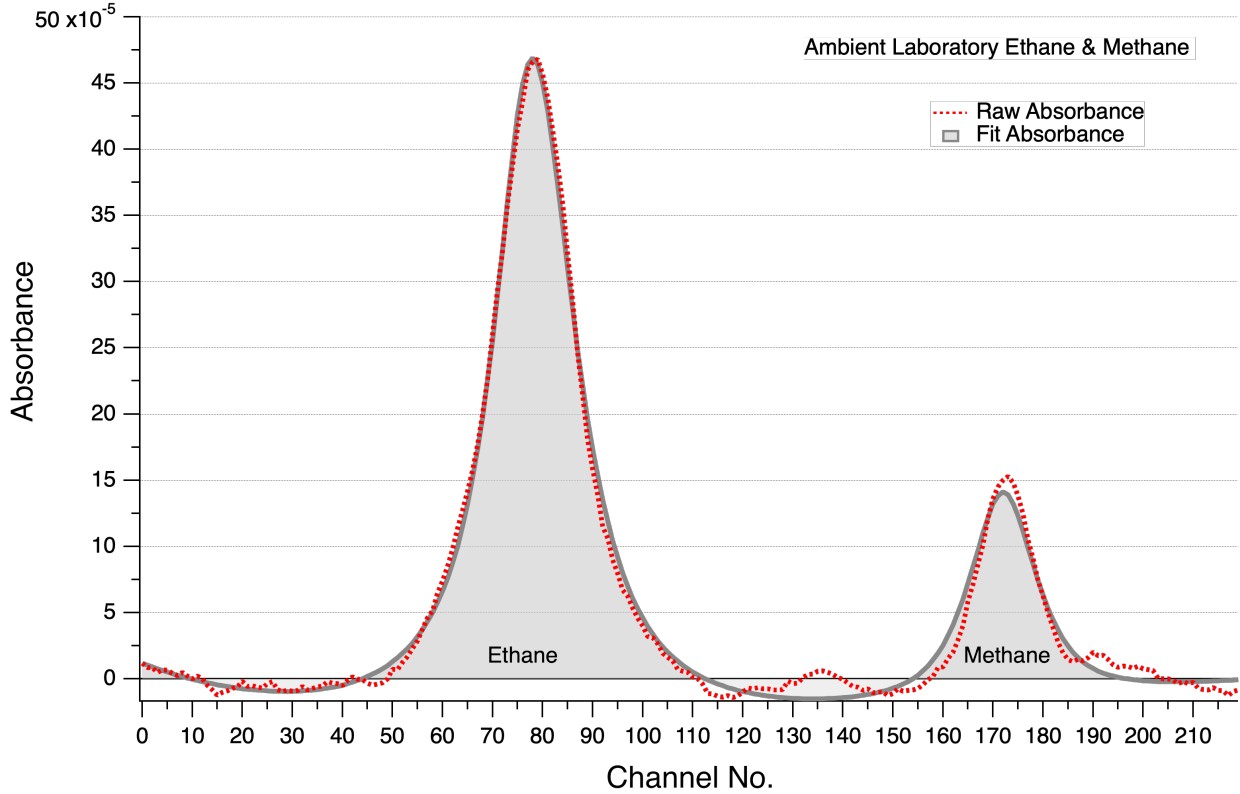

**Figure 6: Ambient ethane and methane raw and fit spectra acquired by sampling laboratory air. The spectra are in channel numbers. The ethane fits out to 4.23 ± 0.025-ppbv, while the methane fits out to 1591 ± 30-ppbv. The methane feature underlying the ethane feature shown in the previous figure is still present but not evident in the fits here since this feature almost perfectly overlaps with ethane. The methane feature on the right in this scan is clearly underestimated since the we have not optimized the width of this feature. The very weak methane feature in between ethane and methane also shows up on the raw spectra but is not properly fit here.**



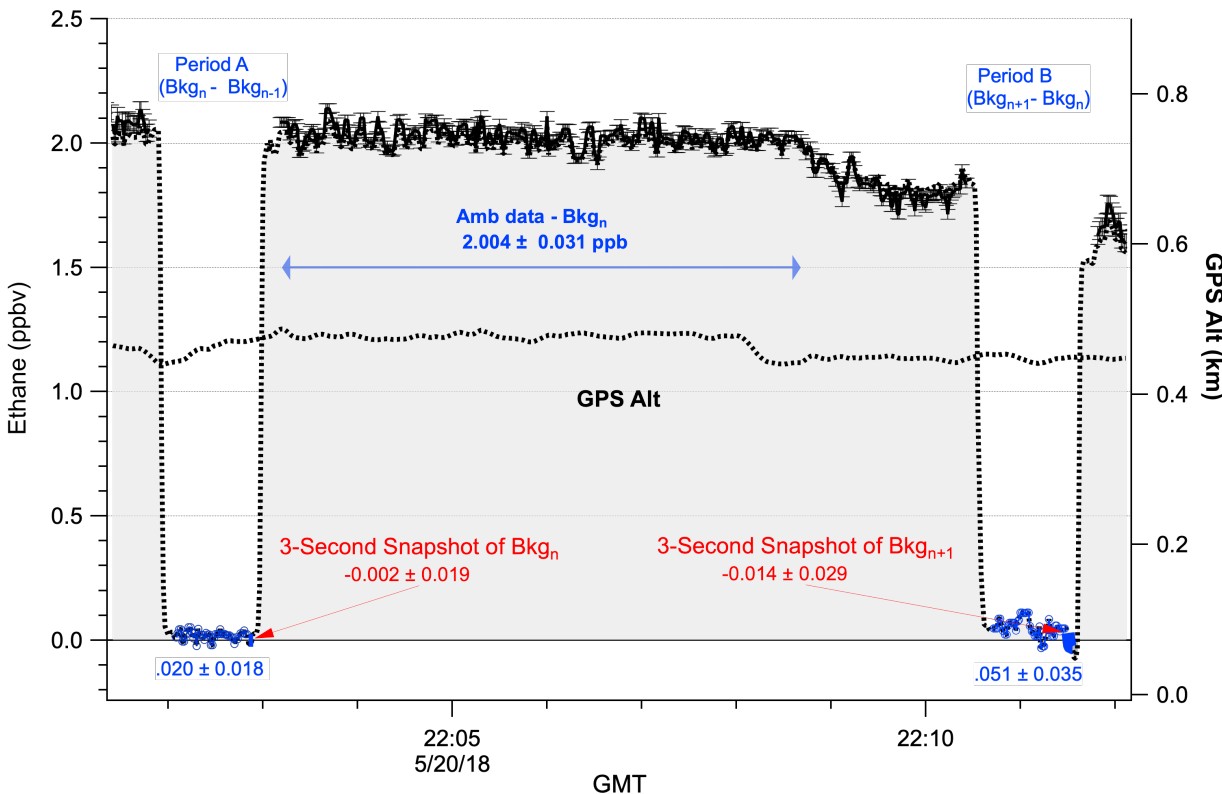

**Figure 7a: Pre- and post- ambient background acquisitions during the 4th field deployment phase. The background values have the same units as the ambient ethane structure (ppbv). The dark blue circles represent the fit of the background difference (Present - Previous). The dark blue line highlighted by the numbers in red represent a 3-second snapshot of the new background-itself. The enclosure pressure change (not shown) over the entire time period here is 0.36 torr. This deployment phase represents the latest improvements where the cell and input/output optics have been further stabilized (see text).**




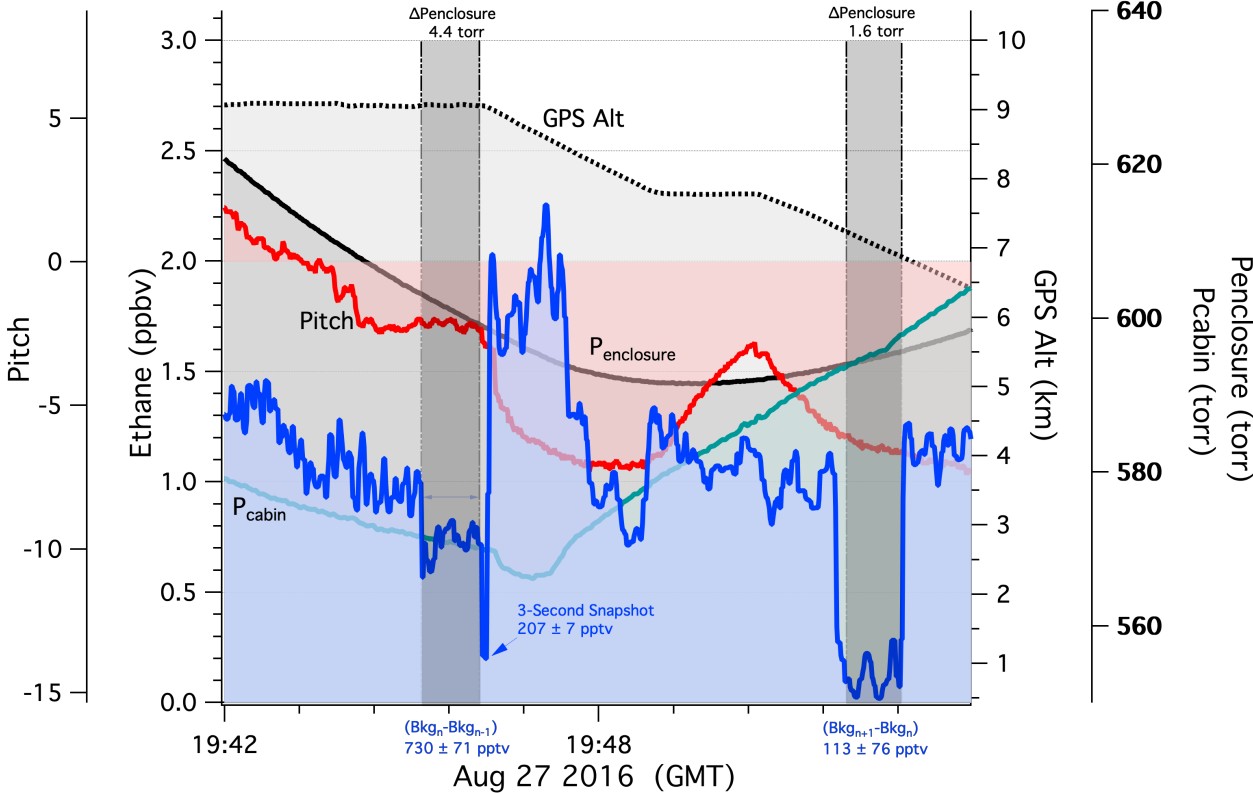

**Figure 7b: Pre- and post- ambient background acquisitions during the 1st field deployment phase in Aug. 27, 2016 before the enclosure was sealed. The blue traces in the shaded regions show background acquisitions before and after the ambient acquisitions. In this case the leaking enclosure caused a pressure change of 4.4-torr in the 1st background region as the cabin pressure changed, causing very dramatic changes not only in the fast noise but also a shift in the background of 0.523 ppb in this case between the 45 second (Bkgn - Bkgn-1) and the (Bkg3sec snapshot - BKgn).**





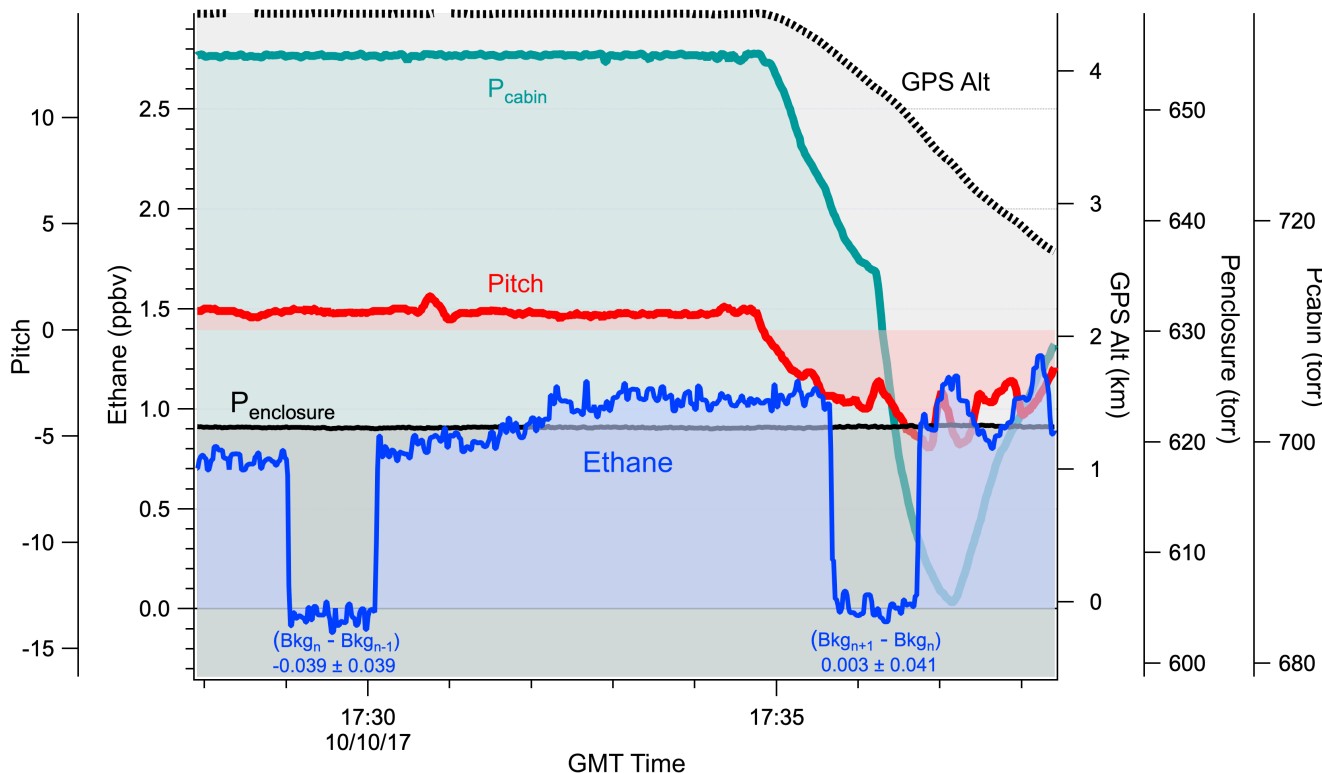

Figure 7c: Pre- and post- ambient background acquisitions during the 3rd field deployment phase in the fall of 2017 in the same format as 7a. As the cabin pressure changes by 37-torr on the descent during the 2nd background period, the enclosure pressure is stable to within 0.23-torr. The optics have not been stabilized here so changes in pitch have a more dramatic effect on the background structure.






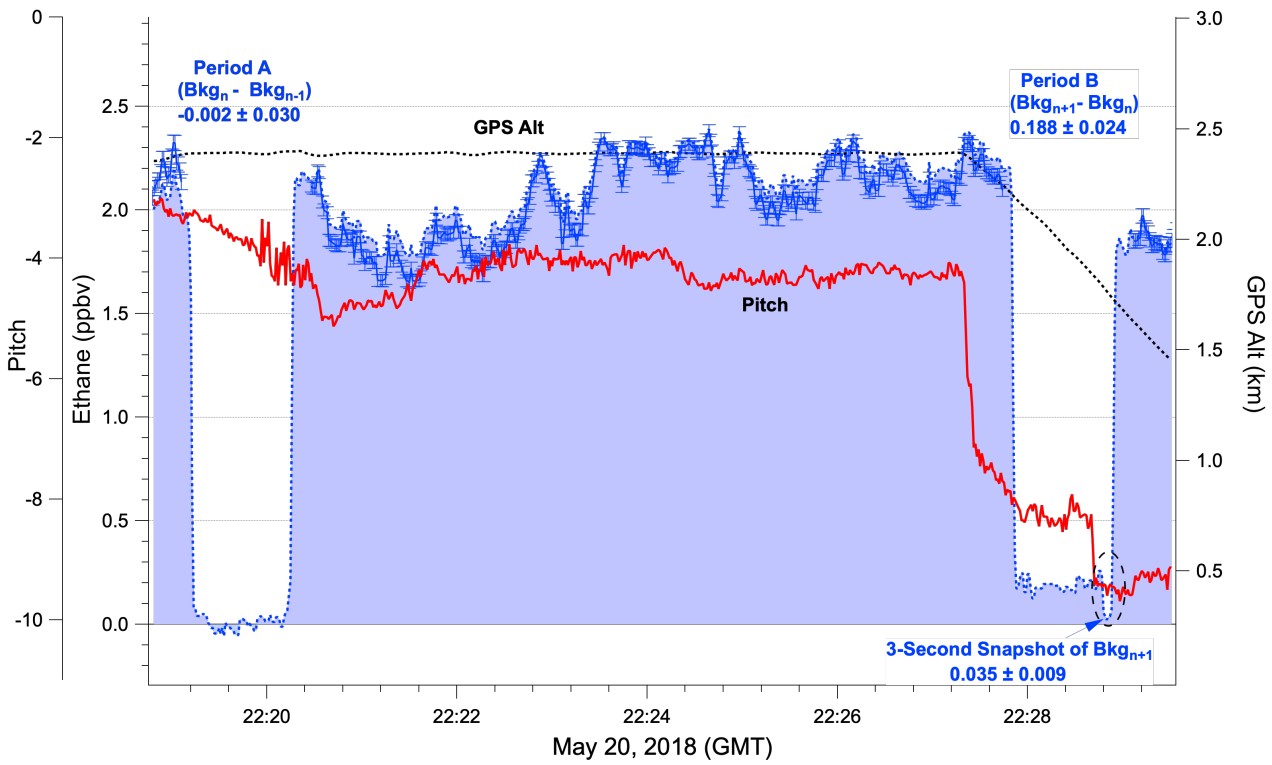

**Figure 7d: Pre- and post- ambient background acquisitions during the 4th field deployment phase. During background period B the aircraft pitch changes causing a small optical change in the background structure which in turn changes the background fit value by ~0.153 ppbv.**








**Figure 8a: Precision histogram of all zero background measurements for 3rd field deployment phase (Oct.3 - Oct.16, 2017), bin width= 0.005 ppbv.**





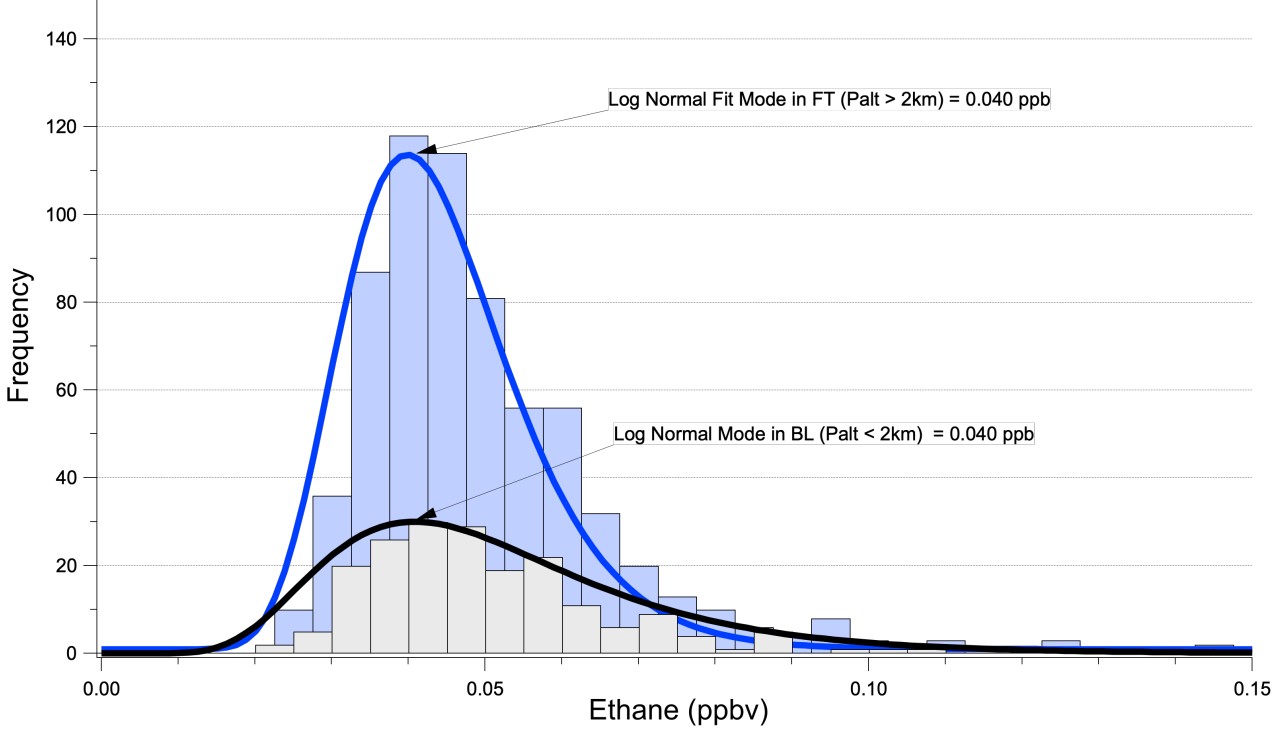

**Figure 8b: Precision histograms of zero air background measurements in PBL and in FT acquired during the Spring 2018 4th field deployment phase. Here the bin width = 0.005 ppbv.**



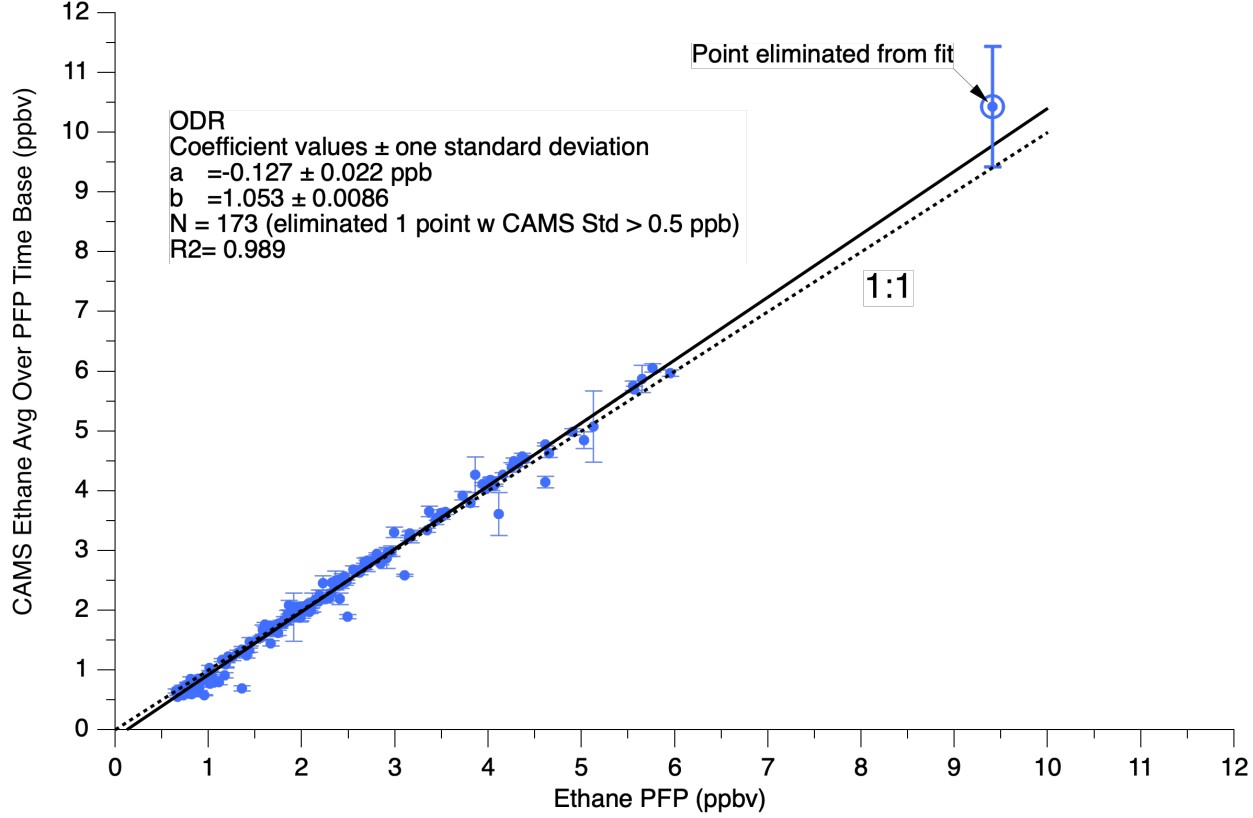

**Figure 9: Spring 2018 IV field deployment phase final comparisons of CAMS average on PFP time base vs PFP.**




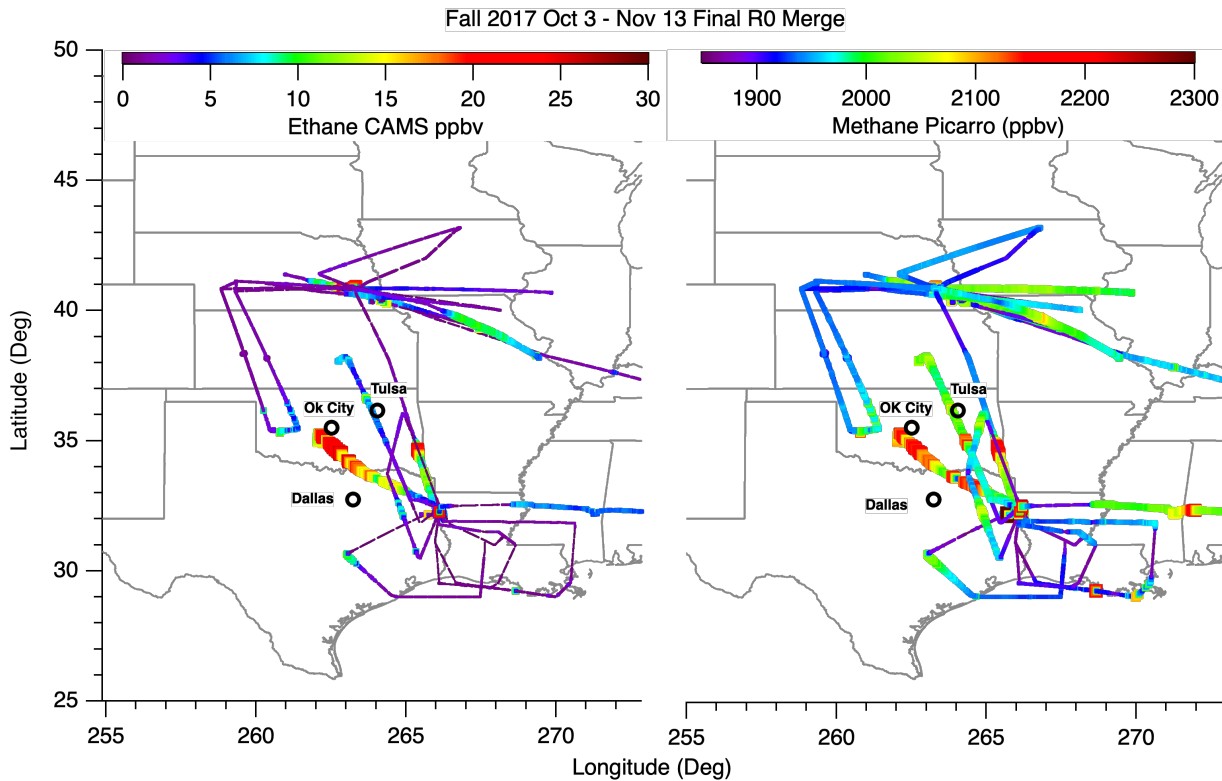

Figure 10: Simultaneous ethane (left plot) and methane (right plot) over the Southeast.

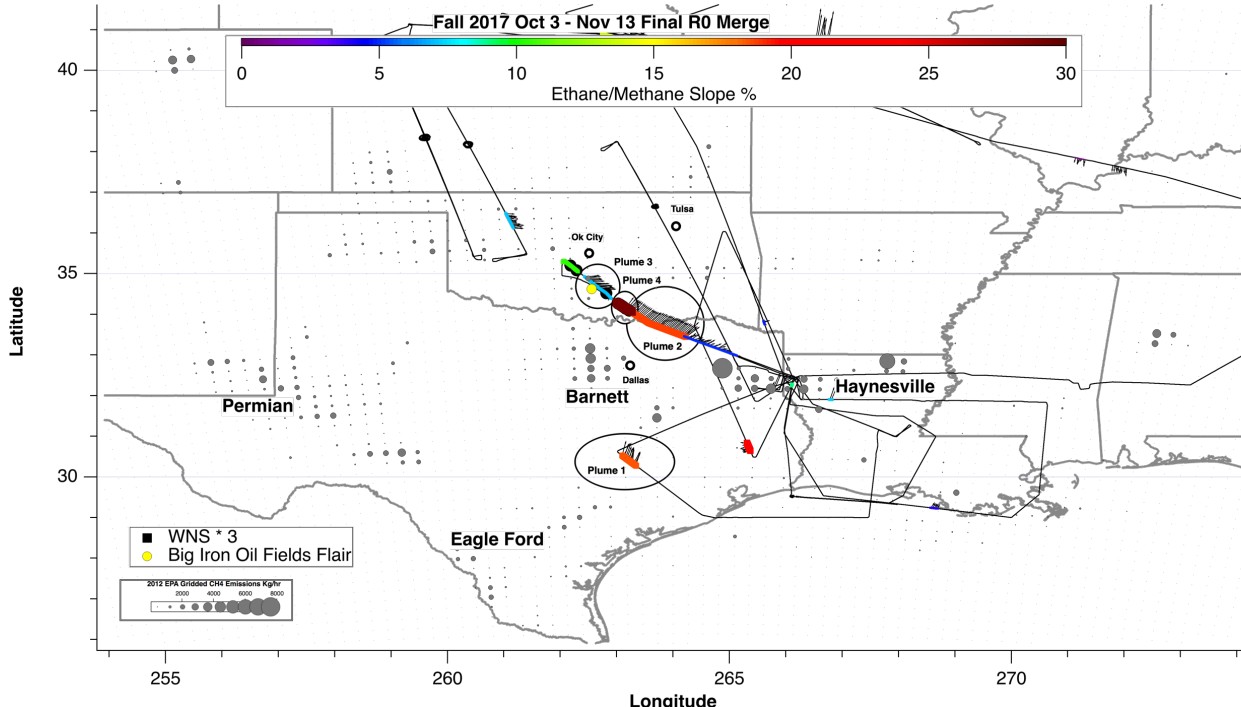

**Figure 11: Ethane/Methane slopes over the Southeast during the Oct-Nov 2017 time period showing 3 plumes with high ethane/methane slopes. The wind directions along the flight tracks are indicated by arrows (the wind speeds, WNS, are multiplied by 3 for emphasis). The 4 major shale plays in this region are indicated along with the 2012 EPA Gridded methane emission rates in the gray filled circles, which are sized by their emission rates.**





**Figure 12a: Ethane-Methane time series plot for plume 1 highlighted in Fig. 11. The CAMS ethane (blue trace) compares well with the PFP ethane measurement shown by the solid gray horizontal points around 20:06. As shown, the ethane-methane slope is 18.4%.**




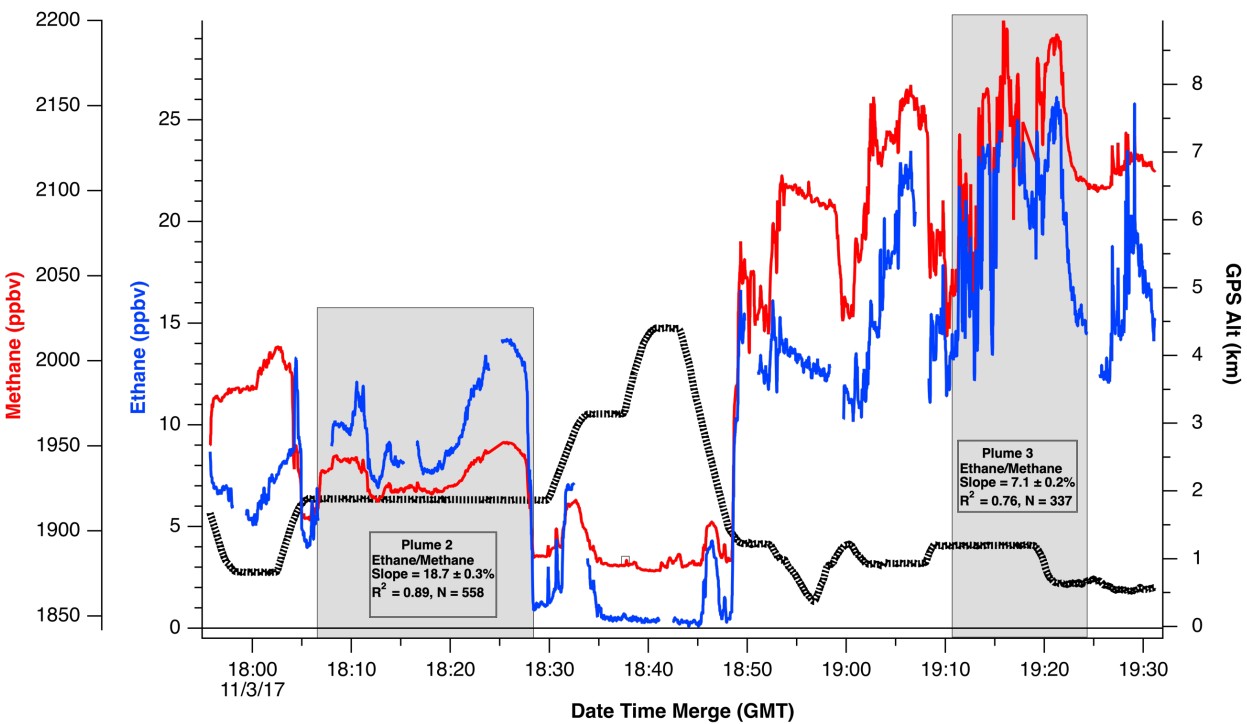

**Figure 12b: Ethane-Methane time series plots for plumes 2 & 3 highlighted in Fig. 11.**



| Parameter | Winter 2017, II | Fall 2017, III | Spring 2018, IV |
|---|---|---|---|
| TMU Average (ppb) | 0.146 | 0.176 | 0.122 |
| TMU Std (ppb) | 0.221 | 0.089 | 0.080 |
| TMU Median (ppb) | 0.095 | 0.164 | 0.098 |
| N | 249923 | 247426 | 331632 |
| TMU Avg % of Ambient | 7.5% | 12.3% | 7.8% |
| TMU Std % of Ambient | 13.9% | 9.1% | 8.4% |
| TMU Median % of Ambient | 4.0% | 9.9% | 5.2% |
| Max. Ambient Ethane Conc. (ppb) | 45.912 | 29.251 | 67.009 |
| Median LOD. (ppb) | 0.051 | 0.051 | 0.044 |


**Table 1: Total Measurement Uncertainty Estimates (TMUs) During All Ambient Ethane Measurements > 0.5-ppb Acquired During 3 of the Campaign Intensives**





| Deployment Phase | Slope | Intercept (ppb) | R₂ | N |
|---|---|---|---|---|
| Winter 2017, II | 1.038 | -0.174 ± 0.241 | 0.998 | 67 |
| Fall 2017, III | 1.044 ± 0.067 | -0.104 ± 0.219 | 0.997 | 96 |
| Spring 2018, IV | 1.053 ± 0.009 | -0.127 ± 0.022 | 0.989 | 173 |
| **Average** | **1.045 ± 0.008** | **-0.135 ± 0.036** | **0.995 ± 0.005** | |


**Table 2: Orthogonal Linear Regressions of the fast CAMS data averaged over the PFP time base vs the PFP data for 3 of the field deployment phases.**
