# Peer review of "Autonomous Airborne Mid-IR Spectrometer for High Precision Measurements of Ethane during the NASA ACT-America Studies"

_Atmospheric Measurement Techniques, 2020_

## Referee Comment (RC1) · Anonymous Referee #1 · 21 Jul 2020

This paper describes the operation and results from an ethane spectrometer. Overall the instrument appears to be state-of-the-art with superb sensitivty and the results are well described. I recommend the manuscript be published after addressing my one major comment and numerous smaller comments.

Major comment:

The overall procedures for determining the background variability are robust and well described, and I especially appreciate the histogram shown in figure 8a. I am concerned, however, that the authors equate precision with the limit of detection (LOD). While the detection limit is determined by the noise of the measurements (and thus

directly connected to the precision), presumably it is also affected by the uncertainty introduced by the need to subtract out the contribution of methane's absorption using data from the PICARRO instrument. At least one equation which defines the LOD (not the same as precision) at a given signal-to-noise ratio should be included. Note that for many other absorption-based measurements the LOD is defined as twice the 1-sigma precision (i.e., signal-to-noise ratio of 2).

Minor comments:

line 34 onward – sentence starting with "There are a . . ." has multiple grammatical errors: 1. remove colon 2. in the parentheses, do not write "i.e., x, y, z, etc.". Just write "e.g., x, y, and z" 3. write ". . .coal mines, wildfires, ruminants and associated manure, landfills water treatment plants, wetlands, and stagnant water ponds" – leave off "as well as biogenic emissions from" and "to name a few".

"Fast measurements, precisely co-aligned in time to remove temporal instrument differences, results in highly correlated emission ratios" 1. should be "result" not "results". 2. The measurements themselves don't result in highly correlated emission ratios. The measures can result in highly correlated concentration ratios, from which emission ratios can be inferrred.

"By contrast, biogenic methane sources reveal enhanced methane with no enhancements in ethane." awkward. Just "By contrast, biogenic methane sources are usually not also ethane sources"

line 43: "ethane is the longest-lived and most abundant non-methane hydrocarbon" – longest lived, really? In some environments, other non-methane hydrocarbons could be more abundant (e.g., isoprene).

line 44: awkward: "higher than methane-OH"

line 44: I do not follow the logic regarding ethane acting as an indirect greenhouse gas. This needs to be better explained, and with a reference. "Greenhouse" need not be

capitalized.

line 53: should be "DFG-based"

lines 59 and 619: metric units please!

line 66: should be "Aerodyne Research, Inc"

lines 70-71 are awkward: "...systems e.g. Pal et al., (2020), quantification of regional, season fluxes of CO2 (Feng et al., 2019; and Zhou et al., 2020) and CH4 Barkley et al, (2019a,b), and evaluation of the Orbiting Carbon Observatory-2 (OCO-2) satellite Bell et al., (in press)"

line 72: Replace "on average fell in the 80 pptv range" with "were approximately 80 pptv"

line 85: "The cabin pressure effect is endemic to all such spectrometers without optical compartment pressure control" The authors should clarify that "optical compartment pressure" refers not just to the optical cell itself but the entire optical set-up.

line 87: remove the colon!!! Later in the sentence, replace the semicolon with a comma.

line 102: remove "as will be discussed"

Figure 1 - nice photo, but please clarify what's inside the black cylinder. That's the optical compartment I assume?! It's only labeled as "TEC Temperature Control" and "Vibration/Shock Isolated and Pressurized Enclosure"

line 119: remove comma

Overall I would have liked to have seen much better usage of commas and colons. I find it surprising that none of the co-authors objected to their frequent incorrect usage.

line 141: replace "...except for a couple of beam dumps" with "...except for two beam dumps"

line 151: "two orthogonally placed spherical mirrors" insert hyphen between "orthogonally" and "placed". Moreover, I am confused by what that means. The two Herriott cell mirrors are at right angles to each other? That can't be right! Also, how many passes are used for this multi-pass cell and what is the effective path length? The path length (48 m) is only noted in figure 2 but not the text.

lines 205 – 209 – What kind of Teflon - PTFE? PFA? Note that not all fluoropolymers are actually "Teflon" brand (from Dupont/Chemours). Easier to just not use the commonly-used word "teflon" and describe what it actually is!

line 210: remove comma

line 215 – this paragraph seems out of place. I recommend placing it after the sampling train and calibrations are described.

line 244 "Calibration standards are measured before and after each flight" The wording is a bit confusing – it could be interpreted to mean that the calibration standard cylinder was measured with something else before and after flights. Perhaps "Calibrations were performed before and after each flight" instead.

line 322: not "times the" but "multiplied by..."

lines 330 onward: I highly recommend using single-letter variables, e.g. P for path length rather than PL.

line 360: "we introduce known C2H6/CH4 calibration standards in compressed gas cylinders from Scott Marrin into the inlet before and after each flight. CAMS direct absorption measurements retrieved ethane mixing ratios that were too low by 6% and all ACT-America data have been subsequently raised by this number" awkward sentences. Raised by what number? Were all the measurements lower than the standards by 6%? Or just some? Please clarify.

line 374: "are accurate on average to within $\pm$ 6% range" please clarify if you are referring to 1 or 2 sigma accuracy

Figure 6, caption: "The ethane fits out to 4.23 ± 0.025-ppbv," awkward language ("fits out"). Perhaps "The fit indicates an ethane mixing ratio of 4.23 . . ."?

line 393 and Figure 7a: "In the case of Period A, we show the residual fit of Bkgn acquired during this period minus Bkgn-1, acquired 7 minutes prior (not shown)" This is confusing. In the figure I do not see any "fits" – just concentrations. Should "fit" be replaced with "derived mixing ratio" or something similar? Ethane-methane slopes: personally I'd prefer to not see them expressed as percentages. ie, just 0.184, not 18.4%
* * *

---

## Referee Comment (RC2) · Anonymous Referee #2 · 18 Aug 2020

**General comments**

The paper "Autonomous Airborne Mid-IR Spectrometer for High Precision Measurements of Ethane during the NASA ACT-America Studies" by Weibring et al. discussed recent results from development and deployment of an airborne laser spectrometer measuring Ethane with high precision. The paper is very well written. The results are excellent and well worth publishing. A few minor comments follow below.

**Specific comments**

P. 14 L. 444: It is true that changes in the background may lead to uncorrelated

Ethane/Methane changes. But it is hard to tell what causes the changes in the ratio unless one permanently measures zero air. In practice a change in Ethane/Methan ratio could also point to a different Methane source. The uncertainty estimation later in the manuscript is very good in this regard as it uses frequent airborne zero air measurements to identify and quantify instrument drift.

P. 17 L. 529: I would write out the formula.

P. 19 L. 587 and Fig. 9: Figure 9 shows 5 data points between 0..5 ppm Ethane that are clearly below the regression line, but they have a very similar slope. Are these from the same flight? Perhaps a systematic offset?

Also in Figure 9 you excluded the 10 ppm Ethane data point. Please explain why. The error bars are just slightly larger than another point at 5 ppm, and the bias from the fit is very similar to the other 5 points mentioned above.

**Technical corrections**

P. 7 L. 210: I would rephrase to: "A sample flow rate of ... yields a cell response time of ... ."

P. 14 L.427: should be take off

Fig. 11: The small labels and wind speed are not readable. Neither the legend for EPA emission rates.

Fig. 12a: Increase fonts of inset to same size as major axes.

---

## Author Comment (AC1) · 18 Sep 2020

We would like to thank Anonymous Referee #1 for insightful comments and suggestions to improve the manuscript. Please see specific responses below.

Reviewer 1

**Major comment:**

The overall procedures for determining the background variability are robust and well described, and I especially appreciate the histogram shown in figure 8a. I am concerned, however, that the authors equate precision with the limit of detection (LOD). While the detection limit is determined by the noise of the measurements (and thus directly connected to the precision), presumably it is also affected by the uncertainty introduced by the need to subtract out the contribution of methane's absorption using data from the PICARRO instrument. At least one equation which defines the LOD (not the same as precision) at a given signal-to-noise ratio should be included. Note that for many other absorption-based measurements the LOD is defined as twice the 1-sigmaprecision (i.e., signal-to-noise ratio of 2).

We agree the reviewer's comments regarding limits of detection and we have replaced the term "LOD" with "precision" throughout the article. The term "TMU" Total Measurement Uncertainty (Lines 530-550) includes the PICCARO measurement uncertainty among other factors (please see modified paragraph and new Equation 3 below). The TMU is defined as 1-sigma.

"As previously stated, the measurement precisions only reveal part of the performance story as changes in background structure acquired during zeroing between ambient acquisitions dictates the overall total measurement uncertainty (TMU). The TMU at the  $1-\sigma$  level is comprised of 5-terms:

$$TMU = \sqrt{A^2 + B^2 + C^2 + D^2 + E^2} \tag{3}$$

These terms are: A) the background precisions prior to each ambient acquisition period; B) temporal changes in the background differences over the course of each ambient acquisition, as discussed in the previous section; C) the uncertainty in the methane interference correction (0.342-ppbv/2000 ppbv [CH4  $\pm$  0.006]), as determined in the laboratory; D) the PICARRO methane measurement error ( $\pm$  1-ppbv x 0.342/2000 =  $\pm$  0.0002 ppbv, https://doi.org/10.3334/ORNLDAAC/1556); and E) the uncertainty in the fitting correction factor employing the input calibration standards."

**Minor Comments:**

line 34 onward – sentence starting with "There are a..." has multiple grammatical errors: 1. remove colon 2. in the parentheses, do not write "i.e., x, y, z, etc.". Just write "e.g., x, y, and z" 3. write "...coal mines, wildfires, ruminants and associated manure, landfills water treatment plants, wetlands, and stagnant water ponds" – leave off "as well as biogenic emissions from" and "to name a few".

Corrected multiple awkward sentences as well as comments regarding lines 43-44 (see below for specific response).

**The paragraph now reads**

"The Atmospheric Carbon and Transport-America (ACT-America) campaign was a four year study composed of five different aircraft campaigns over the continental U.S. to quantify sources, sinks and transport of carbon dioxide (CO2) and methane (CH4), two of the major greenhouse gases. There are a multitude of sources of methane emitting into the atmosphere, such as oil & natural gas exploration and production (e.g., emissions from drilling, on-site processing, storage, flaring and transmission), coal mines, wildfires, ruminants and associated manure, landfills, water treatment plants, wetlands, and stagnant water ponds. In order to evaluate their respective contribution of total emissions, it is important to distinguish and quantify these various sources. One method that has successfully been employed is to utilize fast simultaneous measurements of CH4 with ethane (C2H6). Both gases are co-emitted from oil & natural gas production in varying amounts depending upon the particular shale formation and specific production activity. By contrast, biogenic methane sources are usually not also ethane sources. In addition to its role in characterizing methane sources, ethane is long lived and one of the most abundant non-methane hydrocarbons. Since its reaction rate with OH at 298-K, large enhancements in ethane relative to methane can dramatically affect local OH levels, and hence ethane acts as an indirect greenhouse gas (Kort et al., 2016). This

paper discusses the development and deployment of a precise, accurate, and fast instrument that can reliably measure ethane on small low-flying aircraft and provide invaluable information related to greenhouse emissions."

line 43: "ethane is the longest-lived and most abundant non-methane hydrocarbon" –longest lived, really? In some environments, other non-methane hydrocarbons could be more abundant (e.g., isoprene).

Typically ethane has much higher concentration than other non-methane hydrocarbons, but as pointed out by the reviewer, isoprene concentrations can under certain conditions get higher. The sentence is modified from "ethane is the longest..." to "ethane is long lived..." to account for this.

**line 44: awkward: "higher than methane-OH"**

Re- worded "...with OH is  $\sim$  40 times higher than methane reaction rate with OH" and "...dramatically affect OH levels.." to "...dramatically affect local OH levels.."

**line 44: I do not follow the logic regarding ethane acting as an indirect greenhouse gas. This needs to be better explained, and with a reference. "Greenhouse" need not be capitalized.**

At the local level changes to the OH concentration by ethane will affect the lifetime of other gases (some of which are greenhouse gases). A reference (Kort et al, 2015) has been added.

line 53: should be "DFG-based "DFG based " changed to "DFG-based"

lines 59 and 619: metric units please! Units changed from lbs to kg

line 66: should be "Aerodyne Research, Inc" Changed to "Aerodyne Research, Inc"

lines 70-71 are awkward: "...systems e.g. Pal et al., (2020), quantification of regional,season fluxes of CO2 (Feng et al., 2019; and Zhou et al., 2020) and CH4 Barkley et al.(2019a,b), and evaluation of the Orbiting Carbon Observatory-2 (OCO-2) satellite Bellet al., (in press) Awkward sentences re-worded (see below)

line 72: Replace "on average fell in the 80 pptv range" with "were approximately 80pptv" Replaced "on average fell ..." with "were approximately 80 pptv ..."

**This section now reads**

Yacovitch et al. (2014), Smith et al. (2015), and most recently Kostinek et al. (2019) reported the use of a smaller and lighter weight high performance IR laser system from Aerodyne Research, Inc. and successfully recorded high quality and fast ethane measurements. The latter paper describes improvements to such systems for high performance measurements of CH4, CO2, CO, N2O in addition to C2H6 on the NASA C-130 aircraft during ACT-America. Both the C-130 and B200 were deployed with similar payloads and coordinated flight paths to study the transport of greenhouse gases, primely CO2 and CH4, by mid latitude weather systems. Papers describing these activities are Pal et al., (2020), Feng et al., (2019), Zhou et al., (2020), Barkley et al, (2019a,b), and Bell et al., (in press). Typical airborne ethane measurement precisions reported by Yacovitch et al. (2014) and Smith et al. (2015) were approximately 80 pptv, which is about a factor of 4 higher than when the aircraft was on the ground.

line 85: "The cabin pressure effect is endemic to all such spectrometers without optical compartment pressure control" The authors should clarify that "optical compartment pressure" refers not just to the optical cell itself but the entire optical set-up.

Added clarification of pressure controlled compartment by "The cabin pressure effect is endemic to all such spectrometers without pressure control of the entire optical set-up"

line 87: remove the colon!!! Later in the sentence, replace the semicolon with a comma Removed and replaced comma and semicolon

line 102: remove "as will be discussed"

Removed "...as will be discussed"

Figure 1 - nice photo, but please clarify what's inside the black cylinder. That's the optical compartment I assume?! It's only labeled as "TEC Temperature Control" and "Vibration/Shock Isolated and Pressurized Enclosure Figure label re-worded to clarify "Vibration/Shock isolated and pressurized optical compartment"

Line 119: remove comma Overall I would have liked to have seen much better usage of commas and colons. I find it surprising that none of the co-authors objected to their frequent incorrect usage. Comma removed

line 141: replace "...except for a couple of beam dumps" with "...except for two beam dumps" Replaced with "...except for two beam dumps"

line 151: "two orthogonally placed spherical mirrors" insert hyphen between "orthogonally" and "placed". Moreover, I am confused by what that means. The two Herriott cell mirrors are at right angles to each other? That can't be right! Also, how many passes are used for this multi-pass cell and what is the effective path length? The path length (48 m) is only noted in figure 2 but not the text.

The mirrors are parallel and the re-worded section now reads

"...Similar to the patented multi-pass cell design employed in CAMS-1 (Richter et al., 2015) the present MP offers long path length (47.6 m, 49 roundtrips) and smaller sampling volume ( $\sim$  1 liter) than traditional Herriott cells. This is accomplished employing a sealed hollow core tube in addition to an outer cylindrical tube that provides a vacuumtight optical sampling cell. The inner tube is mounted centered to the cell's longitudinal optical axis, reducing the sampling volume between the two spherical mirrors of a traditional Herriott cell. Its diameter is limited to a radius that provides sufficient clearance of the recirculating beams between the two spherical mirrors..."

lines 205 – 209 – What kind of Teflon - PTFE? PFA? Note that not all fluoropolymers areactually "Teflon" brand (from Dupont/Chemours). Easier to just not use the commonly-used word "teflon" and describe what it actually is! Replaced brand name with "...teflon (PTFE)..".

**line 210: remove comma Re-worded awkward sentence to "For a typical sample flow rate of 4 slm we achieve a cell resonance time of $\sim$ 1 s (1/e)"**

line 215 - this paragraph seems out of place. I recommend placing it after the sampling train and calibrations are described

Paragraph moved to end of section 2.4

line 244 "Calibration standards are measured before and after each flight" The wording is a bit confusing – it could be interpreted to mean that the calibration standard cylinder was measured with something else before and after flights. Perhaps "Calibrations were performed before and after each flight" instead.

The instrument is not "calibrated" as it's a first principle/absolute measurement technique. The calibration standards are used to verify that the instrument absolute accuracy. The section is re-worded as below to clarify.

"...During the Calibration Standard mode, a known mixing ratio  $C_2H_6/CH_4$  (20/ 2000 ppbv) is fed into the zero air stream by a flow controller, which is then added to the inlet. This was performed before and after each flight to verify instrument accuracy. During Ambient and Background modes, a small suck-back flow (0.3-slm) is engaged to draw away any residual standard trapped in the addition line..."

line 322: not "times the" but "multiplied by..." Replaced with "…multiplied by"

lines 330 onward: I highly recommend using single-letter variables, e.g. P for pathlength rather than PL. Replaced double letter variables with single letter

line 360: "we introduce known C2H6/CH4 calibration standards in compressed gas cylinders from Scott Marrin into the inlet before and after each flight. CAMS direct ab-sorption measurements retrieved ethane mixing ratios that were too low by 6% and all ACT-America data have been subsequently raised by this number" awkward sentences. Raised by what number? Were all the measurements lower than the standards by 6%?Or just some? Please clarify. We agree with the reviewer that the calibration section was confusing and we re-worded the entire section (see below) to clarify and reflect the composite of all calibrations. Please note that an error was discovered by Reviewer 2 and the absolute numbers (not the conclusions) have changed slightly after re-processing all data affecting figure 9 as well as

"As indicated, known calibration mixtures of ethane/methane diluted in zero air from a set of working Scott Marrin standards were introduced into the inlet (20 and 2000 ppb) before and after each flight to further validate the direct absorption retrieved values and the fitting approach implemented. Typically, the retrieved ethane values for the working standards were lower than the expected input values based on the manufacturer assigned values times the measured dilution ratio. All reported ambient ethane data were thus based on direct absorption values corrected by the daily working standard correction factors (Assigned cylinder mixing ratio/retrieved values during pre-and post-flight calibrations). Since this procedure relied upon the accuracy of the Scott Marrin working standards, we also verified these standards in the laboratory based on multiple direct absorption measurements employing the CAMS-1 and CAMS-2 instruments. In addition, prior to the 5th field deployment we measured the mixing ratios of various additional ethane standards by direct absorption. These standards included: 1) a gravimetrically prepared ethane/air standard (nominal 5 ppm) from the Apel-Reimer Corporation, which in turn was evaluated by Reimer against NIST Standard Reference Material (SRM) gases; and 2) two additional ethane standards in the 0.3 and 3 ppb range employing Niwot Ridge air prepared and analyzed by the NOAA/ESRL Global Monitoring Division and subsequently analyzed by Detlev Helmig's Atmospheric Research Laboratory at the University of Colorado Institute of Arctic and Alpine Research using standards tied to the Global Greenhouse Gas Reference Network (see for example Helmig et al., 2016). The latter two standards were measured in our laboratory by direct absorption without dilution. Collectively, all the ethane standards comparisons resulted in agreement between our direct absorption values and the assigned cylinder values in the range between -1.2% and +4.8%. It is important to note that the NOAA standards were used by Baier et al. (2019) in their programmable flask package (PFP) ethane measurements. Figure 9a shows an Orthogonal Distance Regression (ODR) linear regression plot of our direct absorption results (with the daily corrections applied) integrated over the PFP time base as a function of the PFP ethane results for the entire spring 2018 4th deployment. Additional ambient ethane ODR comparisons for the 2nd through the 4th field campaigns are provided in Table 2, and these results show agreement between CAMS and the PFP to within 3%. Collectively, all ethane comparisons (ambient and cylinder standard measurements) show agreement within the  $\pm 6\%$  (1 $\sigma$ ) Harrison (2010) cross section uncertainty value."

line 374: "are accurate on average to within  $\pm 6\%$  range" please clarify if you are referring to 1 or 2 sigma accuracy The accuracy is 1 sigma and it is clarified in the re-written section (see above)

Figure 6, caption: "The ethane fits out to 4.23±0.025-ppbv," awkward language ("fitsout"). Perhaps "The fit indicates an ethane mixing ratio of 4.23..."?

Caption sentence is re-worded "...The spectra are in channel numbers. The fit indicates an ethane mixing ratio of  $4.23 \pm 0.025$ -ppbv, while the methane corresponds to  $1591 \pm 30$ -ppbv...."

line 393 and Figure 7a: "In the case of Period A, we show the residual fit of Bkgn acquired during this period minus Bkgn-1, acquired 7 minutes prior (not shown)" This is confusing. In the figure I do not see any "fits" – just concentrations. Should "fit" be replaced with "derived mixing ratio" or something similar? Ethane-methane slopes: personally I'd prefer to not see them expressed as percentages. ie, just 0.184, not18.4%

To clarify that the concentration is based on the residual fit of two consecutive backgrounds the section is re-written as below "... In the case of Period A, we show the derived mixing ratio results from the residual fit of Bkgn acquired during this period minus Bkgn-1, acquired 7 minutes prior (not shown). As illustrated, the fit of the resulting background difference (Bkgn - Bkgn-1) yields a stable background difference ( $0.020 \pm 0.018$  ppb) close to zero..."

Figure 7a

table 2.

The sentence in caption text is re-worded as "...The dark blue circles represent the derived mixing ratios of the background difference (Present - Previous)..."

---

## Author Comment (AC2) · 18 Sep 2020

We would like to thank Anonymous Referee #2 for insightful comments and suggestions to improve the manuscript. Please see specific responses below.

Reviewer 2

Specific comments:

P. 14 L. 444: It is true that changes in the background may lead to uncorrelated Ethane/Methane changes. But it is hard to tell what causes the changes in the ratio unless one permanently measures zero air. In practice a change in Ethane/Methane ratio could also point to a different Methane source. The uncertainty estimation later in the manuscript is very good in this regard as it uses frequent airborne zero air measurements to identify and quantify instrument drift The reviewer is correct. Only looking the methane measurements is not enough judge the validity of the measurements. The paragraph is re-worded as follows " ... Obviously, non-linear drifts or jumps in the true background will cause data errors. Our subsequent data analysis using our ethane data flags such time periods, especially where there are large background changes and/or the ethane data shows such an artificial time dependence. The flagged time period are then manually examined for validity..."

**P. 17 L. 529: I would write out the formula**

Added an equation showing the relationship TMU and subcomponents. The section now reads as below.

The TMU at the 1- $\sigma$  level is comprised of 5-terms:

$$TMU = \sqrt{A^2 + B^2 + C^2 + D^2 + E^2}$$
(3)

These terms are: A) the background precisions prior to each ambient acquisition period; B) temporal changes in the background differences over the course of each ambient acquisition, as discussed in the previous section; C) the uncertainty in the methane interference correction (0.342-ppbv/2000 ppbv [CH4  $\pm$  0.006]), as determined in the laboratory; D) the PICARRO methane measurement error ( $\pm$  1-ppbv x 0.342/2000 =  $\pm$  0.0002 ppbv, https://doi.org/10.3334/ORNLDAAC/1556); and E) the uncertainty in the fitting correction factor employing the input calibration standards."

P. 19 L. 587 and Fig. 9: Figure 9 shows 5 data points between 0..5 ppm Ethane that are clearly below the regression line, but they have a very similar slope. Are these from the same flight? Perhaps a systematic offset? Also in Figure 9 you excluded the 10 ppm Ethane data point. Please explain why. The error bars are just slightly larger than another point at 5 ppm, and the bias from the fit is very similar to the other 5 points mentioned above.

The five points below the correlation line was caused by an intermittent timing problem during the transition between background and ambient measurements that only affected the linear interpolation routines when drift was large during the background acquisition. All data have been re-processed to remove these artifacts. Instead of using the "notch" concentration value as a starting point for the linear background interpolation we instead set the concentration in the beginning of the interpolation to zero regardless of the notch value, which is only used as a diagnostic to flag regions that needs a closer look. The corresponding sections (2.8 and 3.3) have been updated as well as Fig. 9a and Table 2 (see below). Over the 4 mission phases of this study, 95% of the re-processed data resulted in an absolute difference less than 225 pptv and 90% of the data resulted in a change less than 133 pptv. The 10 ppm point is removed by the arbitrary 0.6 - ppbv standard deviation filter designed to catch points where the fill profile of the flask package could compromise the comparison due to rapid changes in the concentration. Please see new fig 9b to better illustrate this. This filter cutoff is higher than our previous, but captures the most egregious point.

Part of Section 2.8

[revised manuscript text omitted]

Technical corrections

P. 7 L. 210: I would rephrase to: "A sample flow rate of ... yields a cell response time of ... . Re-worded the sentence "A sample flow "to "For a typical sample flow rate of 4 slm we achieve a cell resonance time of  $\sim$ 1 s (1/e)"

P. 14 L.427: should be take off Changed "...take..." to "...takeoff..."

Fig. 11: The small labels and wind speed are not readable. Neither the legend for EPA emission rates. The legend has been made bigger as well as wind arrows.

Fig. 12a: Increase fonts of inset to same size as major axes Increased the font size of inset